# GENERATIVE LEARNING WITH EULER PARTICLE TRANSPORT

## ABSTRACT

We propose an Euler particle transport (EPT) approach for generative learning. The proposed approach is motivated by the problem of finding the optimal transport map from a reference distribution to a target distribution characterized by the Monge-Ampere equation. Interpreting the infinitesimal linearization of the Monge-Ampere equation from the perspective of gradient flows in measure spaces leads to a stochastic McKean-Vlasov equation. We use the forward Euler method to solve this equation. The resulting forward Euler map pushes forward a reference distribution to the target. This map is the composition of a sequence of simple residual maps, which are computationally stable and easy to train. The key task in training is the estimation of the density ratios or differences that determine the residual maps. We estimate the density ratios (differences) based on the Bregman divergence with a gradient penalty using deep density-ratio (difference) fitting. We show that the proposed density-ratio (difference) estimators do not suffer from the "curse of dimensionality" if data is supported on a lower-dimensional manifold. Numerical experiments with multi-mode synthetic datasets and comparisons with the existing methods on real benchmark datasets support our theoretical results and demonstrate the effectiveness of the proposed method.

## 1 INTRODUCTION

The ability to efficiently sample from complex distributions plays a key role in a variety of prediction and inference tasks in machine learning and statistics (Salakhutdinov, 2015). The long-standing methodology for learning an underlying distribution relies on an explicit statistical data model, which can be difficult to specify in many applications such as image analysis, computer vision and natural language processing. In contrast, implicit generative models do not assume a specific form of the data distribution, but rather learn a nonlinear map to transform a reference distribution to the target distribution. This modeling approach has been shown to achieve impressive performance in many machine learning tasks (Reed et al., 2016; Zhu et al., 2017). *Generative adversarial networks* (GAN) (Goodfellow et al., 2014), *variational auto-encoders* (VAE) (Kingma & Welling, 2014) and *flow-based methods* (Rezende & Mohamed, 2015) are important representatives of implicit generative models.

In this paper, we propose an Euler particle transport (EPT) approach for learning a generative model by integrating ideas from optimal transport, numerical ODE, density-ratio estimation and deep neural networks. We formulate the problem of generative learning as that of finding a nonlinear transform that pushes forward a reference to the target based on the quadratic Wasserstein distance. Since it is challenging to solve the resulting Monge-Ampère equation, we consider the continuity equation derived from the linearization of the Monge-Ampère equation, which is a gradient flows converging to the target distribution. We solve the Mckean-Vlasov equation associated with the gradient flow using the forward Euler method. The resulting EPT that pushes forward the reference distribution to the target distribution is a composition of a sequence of simple residual maps, which are computationally stable and easy to train. The residual maps are completely determined by the density ratios between the distributions at the current iterations and the target distribution. We estimate density ratios based on the Bregman divergence with a gradient regularizer using deep density-ratio fitting.

We establish bounds on the approximation errors due to linearization of the Monge-Ampère equation, Euler discretization of the Mckean-Vlasov equation, and deep density-ratio estimation. Our result on

the error rate for the proposed density-ratio estimators improves the minimax rate of nonparametric estimation via exploring the low-dimensional structure of the data and circumvents the "curse of dimensionality". Experimental results on multi-mode synthetic data and comparisons with state-of-the-art GANs on benchmark data support our theoretical findings and demonstrate that EPT is computationally more stable and easier to train than GANs. Using simple ReLU ResNets without batch normalization and spectral normalization, we obtained results that are better than or comparable with those using GANs trained with such tricks.

## 2 EULER PARTICLE TRANSPORT

Let $X \in \mathbb{R}^m$ be a random vector with distribution $\nu$, and let $Z$ be a random vector with distribution $\mu$. We assume that $\mu$ has a known and simple form. Our goal is to construct a transformation $\mathcal{T}$ such that $\mathcal{T}_{\#}\mu = \nu$, where $\mathcal{T}_{\#}\mu$ denotes the push-forward distribution of $\mu$ by $\mathcal{T}$, that is, the distribution of $\mathcal{T}(Z)$. Then we can sample from $\nu$ by first generating a $Z \sim \mu$ and calculate $\mathcal{T}(Z)$. In practice, $\nu$ is unknown and only a random sample $\{X_i\}_{i=1}^n$ i.i.d. $\nu$ is available. We must construct $\mathcal{T}$ based on the sample.

There may exist multiple transports $\mathcal{T}$ with $\mathcal{T}_{\#}\mu = \nu$. The optimal transport is the one that minimizes the quadratic Wasserstein distance between $\mu$ and $\nu$ defined by

$$\mathcal{W}_2(\mu, \nu) = \{ \inf_{\gamma \in \Gamma(\mu,\nu)} \mathbb{E}_{(Z,X)\sim\gamma}[\|Z - X\|_2^2] \}^{\frac{1}{2}}, \tag{1}$$

where $\Gamma(\mu, \nu)$ denotes the set of couplings of $(\mu, \nu)$ (Villani, 2008; Ambrosio et al., 2008). Suppose that $\mu$ and $\nu$ have densities $q$ and $p$ with respect to the Lesbeque measure, respectively. Then the optimal transport map $\mathcal{T}$ such that $\mathcal{T}_{\#}\mu = \nu$ is characterized by the Monge-Ampère equation (Brenier, 1991; McCann, 1995; Santambrogio, 2015). Specifically, the minimization problem in (1) admits a unique solution $\gamma = (\mathbb{1}, \mathcal{T})_{\#}\mu$ with $\mathcal{T} = \nabla\Psi, \mu\text{-}a.e.$, where $\mathbb{1}$ is the identity map and $\nabla\Psi$ is the gradient of the potential function $\Psi : \mathbb{R}^m \to \mathbb{R}$. This function is convex and satisfies the Monge-Ampère equation

$$\det(\nabla^2\Psi(\boldsymbol{z})) = \frac{q(\boldsymbol{z})}{p(\nabla\Psi(\boldsymbol{z}))}, \boldsymbol{z} \in \mathbb{R}^m. \tag{2}$$

Therefore, to find the optimal transport $\mathcal{T}$, it suffices to solve (2) for $\Psi$. However, it is challenging to solve this degenerate elliptic equation due to its highly nonlinear nature.

Below we describe the proposed EPT method for obtaining an approximate solution of the Monge-Ampère equation (2). It consists of the following steps: (a) linearizing (2) via residual maps, (b) determining the velocity fields governing the stochastic McKean-Vlasov equation resulting from the linearization, (c) calculating the forward Euler particle transport map and, (d) training the EPT map by estimating the velocity fields from data. Since velocity fields are completely determined by density ratios, this step amounts to nonparametric density ratio estimation. We also provide bounds on the errors due to linearization, discretization and estimation. Mathematical details and proofs are given in the appendix.

**Linearization via residual map** A basic approach to addressing the difficulty due to nonlinearity is linearization. We use a linearization method based on the residual map

$$\mathcal{T}_{t,\Phi_t} = \nabla\Psi = \mathbb{1} + t\nabla\Phi_t, t \geq 0, \tag{3}$$

where $\Phi_t : \mathbb{R}^m \to \mathbb{R}^1$ is a function to be chosen such that the law of $\mathcal{T}_{t,\Phi_t}(Z)$ approaches $\nu$ as $t$ increases (Villani, 2008). We give the specific form of $\Phi_t$ below, see Theorem B.1 in the appendix for details.

This linearization scheme leads to the stochastic process $\mathbf{X}_t : \mathbb{R}^m \to \mathbb{R}^m$ satisfying the McKean-Vlasov equation

$$\frac{\mathrm{d}}{\mathrm{d}t}\mathbf{X}_t(\boldsymbol{x}) = \boldsymbol{v}_t(\mathbf{X}_t(\boldsymbol{x})), \ t \geq 0, \ \text{with} \ \mathbf{X}_0 \sim \mu, \ \mu\text{- a.e.} \ \boldsymbol{x} \in \mathbb{R}^m, \tag{4}$$

where $\boldsymbol{v}_t$ is the velocity vector field of $\mathbf{X}_t$. In addition, we have $\boldsymbol{v}_t = \nabla\Phi_t$. Thus $\boldsymbol{v}_t$ also determines the residual map (3). The details of the derivation are given in Theorems B.2 and B.1. in the appendix. Therefore, estimating the residual map (3) is equivalent to estimating $\boldsymbol{v}_t$.

The movement of $\mathbf{X}_t$ along $t$ is completely governed by $\boldsymbol{v}_t$, given the initial value. We choose a $\boldsymbol{v}_t$ to decrease the discrepancy between the distribution of $\mathbf{X}_t$, say $\mu_t$, at time $t$ and the target $\nu$ with respect to a properly chosen measure. An equivalent formulation of (4) is through the gradient flow $\{\mu_t\}_{t \geq 0}$ with $\{\boldsymbol{v}_t\}_{t \geq 0}$ as its velocity fields, see Proposition B.1 in the appendix. Computationally it is more convenient to work with (4).

**Determining velocity field** The basic intuition is that we should move in the direction that decreases the differences between $\mu_t$ and the target $\nu$. We use an energy functional $\mathcal{L}[\mu_t]$ to measure such differences. An important energy functional $\mathcal{L}[\mu_t]$ is the $f$-divergence (Ali & Silvey, 1966),

$$\mathcal{L}[\mu_t] = \mathbb{D}_f(\mu_t \| \nu) = \int_{\mathbb{R}^m} p(\boldsymbol{x}) f\left(\frac{q_t(\boldsymbol{x})}{p(\boldsymbol{x})}\right) \mathrm{d}\boldsymbol{x}, \tag{5}$$

where $q_t$ is the density of $\mu_t$, $p$ is the density of $\nu$ and $f : \mathbb{R}^+ \to \mathbb{R}$ is assumed to be a twice-differentiable convex function with $f(1) = 0$. We choose $\Phi_t$ such that $\mathcal{L}[\mu_t]$ is minimized. We show in Theorem B.1 in the appendix that $\Phi_t(\boldsymbol{x}) = -f'(r_t(\boldsymbol{x}))$ and $\boldsymbol{v}_t(\boldsymbol{x}) = \nabla \Phi_t(\boldsymbol{x})$. Therefore,

$$\boldsymbol{v}_t(\boldsymbol{x}) = -f''(r_t(\boldsymbol{x}))\nabla r_t(\boldsymbol{x}), \text{ where } r_t(\boldsymbol{x}) = \frac{q_t(\boldsymbol{x})}{p(\boldsymbol{x})}, \ \boldsymbol{x} \in \mathbb{R}^m.$$

For example, if we use the $\chi^2$-divergence with $f(c) = (c-1)^2/2$, then $\boldsymbol{v}_t(\boldsymbol{x}) = \nabla r_t(\boldsymbol{x})$ is simply the gradient of the density ratio. Other types of velocity fields can be obtained by using different energy functionals such as the Lebesgue norm of the density difference, i.e., $\mathcal{L}[\mu_t] = \int_{\mathbb{R}^m} |q_t(\boldsymbol{x}) - p(\boldsymbol{x})|^2 \mathrm{d}\boldsymbol{x}$, see Section B.2 for details.

**The forward Euler method** Numerically, we need to discretize the McKean-Vlasov equation (4). Let $s > 0$ be a small step size. We use the forward Euler method defined iteratively by:

$$\mathcal{T}_k = \mathbb{1} + s\boldsymbol{v}_k, \tag{6}$$
$$\mathbf{X}_{k+1} = \mathcal{T}_k(\mathbf{X}_k), \tag{7}$$
$$\mu_{k+1} = (\mathcal{T}_k)_{\#}\mu_k, \tag{8}$$

where $\mathbf{X}_0 \sim \mu$, $\mu_0 = \mu$, $\boldsymbol{v}_k$ is the velocity field at the $k$th step, $k = 0, 1, ..., K$ for some large $K$. The particle process $\{\mathbf{X}_k\}_{k \geq 0}$ is a discretized version of the continuous process $\{\mathbf{X}_t\}_{t \geq 0}$ in (4). The final transport map is the composition of a sequence of simple residual maps $\mathcal{T}_0, \mathcal{T}_1 \ldots, \mathcal{T}_K$, i.e., $\mathcal{T} = \mathcal{T}_K \circ \mathcal{T}_{K-1} \cdots \circ \mathcal{T}_0$. This updating scheme is based on the forward Euler method for solving equation (4). This is the reason we refer to the proposed method as Euler particle transport (EPT).

**Training EPT** When the target $\nu$ is unknown and only a random sample is available, it is natural to learn $\nu$ by first estimating the discrete velocity fields $\boldsymbol{v}_k$ at the sample level and then plugging the estimator of $\boldsymbol{v}_k$ in (6). For example, if we use the $f$-divergence as the energy functional, estimating $\boldsymbol{v}_k(\boldsymbol{x}) = -f''(r_k(\boldsymbol{x}))\nabla r_k(\boldsymbol{x})$ boils down to estimating the density ratios $r_k(\boldsymbol{x}) = q_k(\boldsymbol{x})/p(\boldsymbol{x})$ dynamically at each iteration $k$. Nonparametric density-ratio estimation using Bregman divergences and gradient regularizer are discussed in Section 4 below. Let $\hat{\boldsymbol{v}}_k$ be the estimated velocity fields at the $k$th iteration. The $k$th estimated residual map is $\widehat{\mathcal{T}}_k = \mathbb{1} + s\hat{\boldsymbol{v}}_k$. Finally, the trained map is

$$\widehat{\mathcal{T}} = \widehat{\mathcal{T}}_K \circ \widehat{\mathcal{T}}_{K-1} \circ \cdots \circ \widehat{\mathcal{T}}_0. \tag{9}$$

**Theoretical guarantees** We establish the following bound on the approximation error due to the linearization of the Monge-Ampère equation under appropriate conditions:

$$\mathcal{W}_2(\mu_t, \nu) = \mathcal{O}(e^{-\lambda t}), \tag{10}$$

for some $\lambda > 0$, see Proposition B.1 in the appendix. Therefore, $\mu_t$ converges to $\nu$ exponentially fast as $t \to \infty$. For an integer $K \geq 1$ and a small $s > 0$, let $\{\mu_t^s : t \in [ks, (k+1)s), k = 0, \ldots, K\}$ be a piecewise constant interpolation between $\mu_{ks}$ and $\mu_{(k+1)s}$, $k = 0, 1, \ldots, K$. Under the assumption that the velocity fields $\boldsymbol{v}_t$ are Lipschitz continuous with respect to $(\boldsymbol{x}, \mu_t)$, the discretization error of $\mu_t^s$ can be bounded in a finite time interval $[0, T]$ as follows:

$$\sup_{t \in [0,T)} \mathcal{W}_2(\mu_t, \mu_t^s) = \mathcal{O}(s). \tag{11}$$

The proof of (11) is given in Proposition B.2 in the appendix. The error bounds (10) and (11) imply that the distributions of the particles $\mathbf{X}_k$ generated by the EPT map defined in (7) with a small $s$ and a sufficiently large $k$ converges to the target $\nu$ at the rate of discretization size $s$.

When training the EPT map, we use the deep neural networks to estimate the density ratios (density differences) with samples. In Theorem 4.1, we provide an estimation error bound that improves the minimax rate of deep nonparametric estimation via exploring the low-dimensional structure of data and circumvents the "curse of dimensionality." Thus this result is of independent interest in nonparametric estimation using deep neural networks.

## 3 IMPLEMENTATION

We now described how to implement EPT and train the optimal transport $\mathcal{T}$ with an i.i.d. sample $\{X_i\}_{i=1}^n \subset \mathbb{R}^m$ from an unknown target distribution $\nu$. The EPT map is trained via the forward Euler iteration (6)-(8) with a small step size $s > 0$. The resulting map is a composition of a sequence of residual maps, i.e., $\mathcal{T}_K \circ \mathcal{T}_{K-1} \circ ... \circ \mathcal{T}_0$ for a large $K$. As implied by Theorem 4.1 in Section 4, each $\mathcal{T}_k, k = 0, ..., K$ can be estimated with high accuracy by $\widehat{\mathcal{T}}_k = \mathbb{1} + s\hat{\boldsymbol{v}}_k$, where $\hat{\boldsymbol{v}}_k(\boldsymbol{x}) = -f''(\widehat{R}_\phi(\boldsymbol{x}))\nabla \widehat{R}_\phi(\boldsymbol{x})$. Here $\widehat{R}_\phi$ is the density-ratio estimator defined in (14) below based on $\{Y_i\}_{i=1}^n \sim q_k$ and the data $\{X_i\}_{i=1}^n \sim p$. Therefore, according to the EPT map (9), the particles

$$\widehat{\mathcal{T}}(\tilde{Y}_i) \equiv \widehat{\mathcal{T}}_K \circ \widehat{\mathcal{T}}_{K-1} \circ ... \circ \widehat{\mathcal{T}}_0(\tilde{Y}_i), i = 1, \ldots, n$$

serve as samples drawn from the target distribution $\nu$, where particles $\{\tilde{Y}_i\}_{i=1}^n \subset \mathbb{R}^m$ are sampled from a simple reference distribution $\mu$.

In many applications, high-dimensional complex data such as images, texts and natural languages, tend to have low-dimensional latent features. To learn generative models with latent low-dimensional structures, it is beneficial to have the option of first sampling particles $\{Z_i\}_{i=1}^n$ from a low-dimensional reference distribution $\tilde{\mu} \in \mathcal{P}_2(\mathbb{R}^\ell)$ with $\ell \ll d$. Then we apply $\widehat{\mathcal{T}}$ to particles $\tilde{Y}_i = G_\theta(Z_i), i = 1, ..., n$, where we introduce another deep neural network $G_\theta : \mathbb{R}^\ell \to \mathbb{R}^m$ with parameter $\theta$. We can estimate $G_\theta$ via fitting the pairs $\{(Z_i, \tilde{Y}_i)\}_{i=1}^n$. We describe the EPT algorithm below.

- **Outer loop for modeling low dimensional latent structure (optional)**
    - Sample $\{Z_i\}_{i=1}^n \subset \mathbb{R}^\ell$ from a low-dimensional reference distribution $\tilde{\mu}$ and let $\tilde{Y}_i = G_\theta(Z_i), i = 1, 2, \ldots, n$.
    - **Inner loop for finding the push-forward map**
        * If there are no outer loops, sample $\tilde{Y}_i \sim \mu, i = 1, \ldots, n$.
        * Get $\hat{\boldsymbol{v}}(\boldsymbol{x}) = -f''(\widehat{R}_\phi(\boldsymbol{x}))\nabla \widehat{R}_\phi(\boldsymbol{x})$ via solving (14) below with $Y_i = \tilde{Y}_i$. Set $\widehat{\mathcal{T}} = \mathbb{1} + s\hat{\boldsymbol{v}}$ with a small step size $s$.
        * Update the particles $\tilde{Y}_i = \widehat{\mathcal{T}}(\tilde{Y}_i), i = 1, \ldots, n$.
    - **End inner loop**
    - If there are outer loops, update the parameter $\theta$ of $G_\theta(\cdot)$ via solving $\min_\theta \sum_{i=1}^n \|G_\theta(Z_i) - \tilde{Y}_i\|_2^2/n$.
- **End outer loop**

## 4 DEEP DENSITY-RATIO AND DENSITY-DIFFERENCE FITTING

The evaluation of velocity fields depends on the dynamic estimation of a discrepancy between the push-forward distribution $q_t$ and the target distribution $p$. Density-ratio and density-difference fitting with the Bregman score provides a unified framework for such discrepancy estimation without estimating each density separately (Gneiting & Raftery, 2007; Dawid, 2007; Sugiyama et al., 2012a;b; Kanamori & Sugiyama, 2014).

Let $r(\boldsymbol{x}) = q(\boldsymbol{x})/p(\boldsymbol{x})$ be the density ratio between a given density $q(\boldsymbol{x})$ and the target $p(\boldsymbol{x})$. Let $g : \mathbb{R} \to \mathbb{R}$ be a differentiable and strictly convex function. The separable Bregman score with the base probability density $p$ for measuring the discrepancy between $r$ and a measurable function $R : \mathbb{R}^m \to \mathbb{R}^1$ is

$$\mathfrak{B}(r, R) = \mathbb{E}_{X \sim p}[g'(R(X))R(X) - g(R(X))] - \mathbb{E}_{X \sim q}[g'(R(X))].$$

Here we focus on the widely used least-squares density-ratio (LSDR) fitting with $g(c) = (c-1)^2$ as a working example, i.e.,

$$\mathfrak{B}_{\text{LSDR}}(r, R) = \mathbb{E}_{X \sim p}[R(X)^2] - 2\mathbb{E}_{X \sim q}[R(X)] + 1. \tag{12}$$

For other choice of $g$, such as $g(c) = c \log c - (c+1) \log(c+1)$ corresponding to estimating $r$ via the logistic regression (LR), and the scenario of density difference fitting will be presented in detail in Section B.3.1.

**Gradient regularizer**  The distributions of real data may have a low-dimensional structure with their support concentrated on a low-dimensional manifold, which may cause the $f$-divergence to be ill-posed due to non-overlapping supports. To exploit such underlying low-dimensional structures and avoid ill-posedness, we derive a simple weighted gradient regularizer $\frac{1}{2}\mathbb{E}_p[g''(R)\|\nabla R\|_2^2]$, motivated by recent works on smoothing via noise injection (Sønderby et al., 2017; Arjovsky & Bottou, 2017). This serves as a regularizer for deep density-ratio fitting. For example, with $g(c) = (c-1)^2$, the resulting gradient regularizer is

$$\mathbb{E}_p[\|\nabla R\|_2^2], \tag{13}$$

which recovers the well-known squared Sobolev semi-norm in nonparametric statistics. Gradient regularization stabilizes and improves the long time performance of EPT. The detailed derivation is presented in Section B.3.2.

**LSDR estimation with gradient regularizer**  Let $\{X_i\}_{i=1}^n$ and $\{Y_i\}_{i=1}^n$ be two collections of i.i.d data from densities $p(\boldsymbol{x})$ and $q(\boldsymbol{x})$, respectively. Let $\mathcal{H} \equiv \mathcal{H}_{\mathcal{D},\mathcal{W},\mathcal{S},\mathcal{B}}$ be the set of ReLU neural networks $R_\phi$ with parameter $\phi$, depth $\mathcal{D}$, width $\mathcal{W}$, size $\mathcal{S}$, and $\|R_\phi\|_\infty \leq \mathcal{B}$. We combine the least squares loss (12) with the gradient regularizer (13) as our objective function. The resulting gradient regularized LSDR estimator of $r = p/q$ is given by

$$\widehat{R}_\phi \in \arg\min_{R_\phi \in \mathcal{H}} \frac{1}{n} \sum_{i=1}^n [R_\phi(X_i)^2 - 2R_\phi(Y_i)] + \alpha \frac{1}{n} \sum_{i=1}^n \|\nabla R_\phi(X_i)\|_2^2, \tag{14}$$

where $\alpha \geq 0$ is a regularization parameter.

**Estimation error bound**  We first show that the density ratio $r$ is identifiable through the objective function by proving that, at the population level, we can recover the density ratio $r$ via minimizing

$$\mathfrak{B}_{\text{LSDR}}^\alpha(R) = \mathfrak{B}_{\text{LSDR}}(r, R) + \alpha \mathbb{E}_p[\|\nabla R\|_2^2] + \mathcal{C},$$

where $\mathfrak{B}_{\text{LSDR}}$ is defined in (12) and $\mathcal{C} = \mathbb{E}_{X \sim q}[r^2(X)] - 1$.

**Lemma 4.1.** *For any $\alpha \geq 0$, we have $r \in \arg\min_R \mathfrak{B}_{\text{LSDR}}^\alpha(R)$. In addition, $\mathfrak{B}_{\text{LSDR}}^\alpha(R) \geq 0$ for any $R$ with $\mathbb{E}_{X \sim p} R^2(X) < \infty$, and $\mathfrak{B}_{\text{LSDR}}^\alpha(R) = 0$ iff $R(\boldsymbol{x}) = r(\boldsymbol{x}) = 1$ $(q, p)$-a.e. $\boldsymbol{x} \in \mathbb{R}^m$.*

This identifiabiity result shows that the target density ratio is the unique minimizer of the population version of the empirical criterion in (14). This provides a the basis for establishing the convergence result of deep nonparametric density-ratio estimation.

Next we bound the nonparametric estimation error $\|\widehat{R}_\phi - r\|_{L^2(\nu)}$ under the assumptions that the support of $\nu$ is concentrated on a compact low-dimensional manifold and $r$ is Lipsichiz continuous. Let $\mathfrak{M} \subseteq [-c, c]^m$ be a Riemannian manifold (Lee, 2010) with dimension $\mathfrak{m}$, condition number $1/\tau$, volume $\mathcal{V}$, geodesic covering regularity $\mathcal{R}$, and $\mathfrak{m} \ll \mathcal{M} = \mathcal{O}(\mathfrak{m} \ln(m\mathcal{V}\mathcal{R}/\tau)) \ll m$. Denote $\mathfrak{M}_\epsilon = \{\boldsymbol{x} \in [-c, c]^m : \inf\{\|\boldsymbol{x} - \boldsymbol{y}\|_2 : \boldsymbol{y} \in \mathfrak{M}\} \leq \epsilon\}$, $\epsilon \in (0, 1)$.

**Theorem 4.1.** *Assume $\text{supp}(r) = \mathfrak{M}_\epsilon$ and $r(\boldsymbol{x})$ satisfies $|r(\boldsymbol{x})| \leq B$ for a finite constant $B > 0$ and is Lipschitz continuous with Lipschitz constant $L$. Suppose the topological parameter of $\mathcal{H}_{\mathcal{D},\mathcal{W},\mathcal{S},\mathcal{B}}$ in (14) with $\alpha = 0$ satisfies $\mathcal{D} = \mathcal{O}(\log n)$, $\mathcal{W} = \mathcal{O}(n^{\frac{\mathcal{M}}{2(2+\mathcal{M})}}/\log n)$, $\mathcal{S} = \mathcal{O}(n^{\frac{\mathcal{M}-2}{\mathcal{M}+2}}/\log^4 n)$, and $\mathcal{B} = 2B$. Then,*

$$\mathbb{E}_{\{X_i, Y_i\}_{i=1}^n}[\|\widehat{R}_\phi - r\|_{L^2(\nu)}^2] \leq C(B^2 + cLm\mathcal{M})n^{-2/(2+\mathcal{M})},$$

*where $C$ is a universal constant.*

The error bound established in Theorem 4.1 for the nonparametric deep density-ratio fitting is new. This result is of independent interest for nonparametric estimation with deep neural networks.

The above derived rate $\mathcal{O}(n^{-\frac{2}{2+\mathcal{M}\ln m}})$ is faster than the optimal rate of convergence for nonparametric estimation of a Lipschitz target in $\mathbb{R}^m$, where the optimal rate is $\mathcal{O}(n^{-\frac{2}{2+m}})$ (Stone, 1982; Schmidt-Hieber, 2020) as long as the intrinsic dimension $\mathcal{M}$ of the data is much smaller than the ambient dimension $m$. Therefore, the proposed density-ratio estimators circumvent the "curse of dimensionality" if data is supported on a lower-dimensional manifold.

## 5 RELATED WORK

We discuss connections between EPT and the existing related works. The existing generative models, such as VAEs, GANs and flow-based methods, parameterize a transform map with a neural network, say $G$, that solves

$$\min_G \mathfrak{D}(G_\# \mu, \nu), \tag{15}$$

where $\mathfrak{D}(\cdot, \cdot)$ is an integral probability discrepancy. The original GAN (Goodfellow et al., 2014), $f$-GAN (Nowozin et al., 2016) and WGAN (Arjovsky et al., 2017) solve the dual form of (15) by parameterizing the dual variable using another neural network with $\mathfrak{D}$ as the JS-divergence, the $f$-divergence and the 1-Wasserstein distance, respectively. Based on the fact that the 1-Wasserstein distance can be evaluated from samples via linear programming (Sriperumbudur et al., 2012), Liu et al. (2018) and Genevay et al. (2018) proposed training the primal form of WGAN via a two-stage method that solves the linear programm. SWGAN (Deshpande et al., 2018) and MMDGAN (Li et al., 2017; Binkowski et al., 2018) use the sliced quadratic Wasserstein distance and the maximum mean discrepancy (MMD) as $\mathfrak{D}$, respectively.

Vanilla VAE (Kingma & Welling, 2014) approximately solves the primal form of (15) with the KL-divergence loss under the framework of variational inference. Several authors have proposed methods that use optimal transport losses, such as various forms of Wasserstein distances between the distribution of learned latent codes and the prior distribution as the regularizer in VAE to improve performance. These methods include WAE (Tolstikhin et al., 2018), Sliced WAE (Kolouri et al., 2019) and Sinkhorn AE (Patrini et al., 2019).

Discrete time flow-based methods minimize (15) with the KL divergence loss (Rezende & Mohamed, 2015; Dinh et al., 2015; 2017; Kingma et al., 2016; Papamakarios et al., 2017; Kingma & Dhariwal, 2018). Grathwohl et al. (2019) proposed an ODE flow approach for fast training in such methods using the adjoint equation (Chen et al., 2018b). By introducing the optimal transport tools into maximum likelihood training, Chen et al. (2018a) and Zhang et al. (2018) considered continuous time flow. Chen et al. (2018a) proposed a gradient flow in measure spaces in the framework of variational inference and then discretized it with the implicit movement minimizing scheme (De Giorgi, 1993; Jordan et al., 1998). Zhang et al. (2018) considered gradient flows in measure spaces with time invariant velocity fields. CFGGAN (Johnson & Zhang, 2018) derived from the perspective of optimization in the functional space is a special form of EPT with $\mathcal{L}[\cdot]$ taken as the KL divergence. SW flow (Liutkus et al., 2019) and MMD flow (Arbel et al., 2019) are gradient flows in measure spaces. MMD flow can be recovered from EPT by first choosing $\mathcal{L}[\cdot]$ as the Lebesgue norm and then projecting the corresponding velocity vector fields onto reproducing kernel Hilbert spaces, please see Appendix B.4 for a proof. However, neither SW flow nor MMD flow can model hidden low-dimensional structure with the particle sampling procedure.

SVGD in (Liu, 2017) and the proposed EPT are both particle methods based on gradient flow in measure spaces. However, the SVGD samples from an un-normalized density, while EPT focuses on generative leaning, i.e., learning the distribution from samples. At the population level, projecting the velocity fields of EPT with KL divergence onto reproducing kernel Hilbert Spaces will recover the velocity fields of SVGD. The proof is given in Appendix B.5. Score-based methods in (Song & Ermon, 2019; 2020; Ho et al., 2020) are also particle methods based on unadjusted Langevin flow and deep score estimators. At the population level, the velocity fields of these score-based methods are random since they have a Brownian motion term, while the velocity fields of EPT are deterministic. At the sample level, these score-based methods need to learn a vector-valued deep score function. while in EPT we need to estimate the density ratios which are scalar functions.

## 6 EXPERIMENTS

The implementation details on numerical settings, network structures, SGD optimizers and hyper-parameters are given in the appendix. All experiments are performed using NVIDIA Tesla K80 GPUs. The PyTorch code of EPT is available at `https://github.com/anonymous/EPT`.

**2D Examples.** We use EPT to learn 2D distributions adapted from Grathwohl et al. (2019) with multiple modes and density ridges. The first row in Figure 1 shows kernel density estimation (KDE) plots of 50k samples from target distributions including (from left to right) *8Gaussians, pinwheel, moons, checkerboard, 2spirals,* and *circles*.

The second and third rows show the KDE plots of learned samples via EPT with $f$-divergence/ Lebesgue norm (left six of the second/ third row), and surface plots of estimated density ratio/ difference after 20k iterations of EPT with $f$-divergence/ Lebesgue norm (right six of the second/ third row), respectively. Clearly, the generated samples via EPT are nearly indistinguishable from those of the target samples and the estimated density-ratio/ difference functions are approximately equal to 1/0, indicating the learnt distribution matches the target well.

We further visualize the transport maps learned with $5squares$ and $large4gaussians$ from $4squares$ and $small4gaussians$, respectively. We use 200 particles connected with grey lines to manifest the learned transport maps. As shown in the left two figures in Figures 2, the central squares of $5squares$ were learned better with the gradient penalty, which is consistent with the result of the estimated density-ratio right two figures in Figure 2. For $large4gaussians$, the learned transport map exhibited some optimality under quadratic Wasserstein distance due to the obvious correspondence between the samples in left two figures in Figure 2.

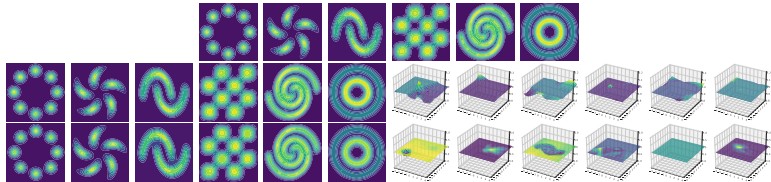

Figure 1: KDE plots of the target samples (the first row), KDE plots of the learned samples via EPT with $f$-divergence/ Lebesgue norm (left six of the second/third row)) and surface plots of estimated density ratio/ difference after 20k iterations of EPT with $f$-divergence/ Lebesgue norm (right six of the second/ third row).

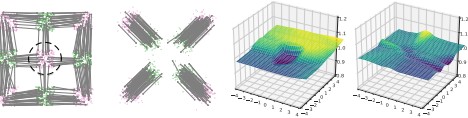

Figure 2: Learned transport maps (left two) and estimated density ratio (right two) in learning $5squares$ from $4squares$ and learning $large4gaussians$ from $small4gaussians$.

**Results on Benchmark Image Data.** We show the performance of applying EPT to benchmark data MNIST (LeCun et al., 1998), CIFAR10 (Krizhevsky & Hinton, 2009) and CelebA (Liu et al., 2015) using ReLU ResNets without batch normalization and spectral normalization. The particle evolutions on MNIST and CIFAR10 without using outer loop are shown in Figure 3. Clearly, EPT can transport samples from a multivariate normal distribution into a target distribution.

We further compare EPT using the outer loop with the generative models including WGAN, SNGAN and MMDGAN. We considered different $f$-divergences, including Pearson's $\chi^2$, KL, JS and logD (Gao et al., 2019)) and different deep density-ratio fitting methods (LSDR and LR). Table 1 shows FID (Heusel et al., 2017) evaluated with five bootstrap sampling of EPT with four divergences on CIFAR10. We can see that EPT using ReLU ResNets without batch normalization and spectral normalization attains (usually better) comparable FID scores with the state-of-the-art generative models. Comparisons of the real samples and learned samples on MNIST, CIFAR10 and CelebA are shown in Figure 4, where high-fidelity learned samples are comparable to real samples visually.

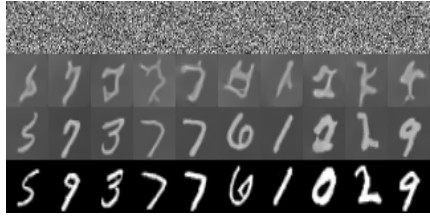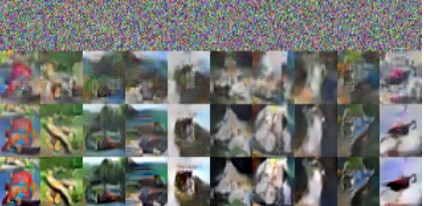

Figure 3: Particle evolution of EPT on MNIST and CIFAR10.

Table 1: Mean (standard deviation) of FID scores on CIFAR10. The FID score of NSCN is reported in Song & Ermon (2019) and results in the right table are adapted from Arbel et al. (2018).

| Models | CIFAR10 (50k) |
|---|---|
| EPT-LSDR-$\chi^2$ | **24.9 (0.1)** |
| EPT-LR-KL | 25.9 (0.1) |
| EPT-LR-JS | 25.3 (0.1) |
| EPT-LR-logD | **24.6 (0.1)** |
| NCSN | 25.3 |

| Models | CIFAR10 (50k) |
|---|---|
| WGAN-GP | 31.1 (0.2) |
| MMDGAN-GP-L2 | 31.4 (0.3) |
| SN-GAN | 26.7 (0.2) |
| SN-SMMDGAN | **25.0 (0.3)** |

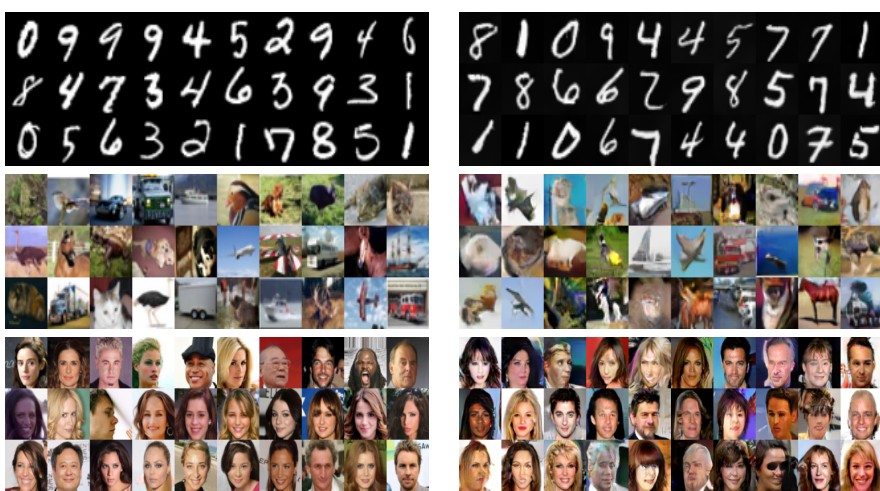

Figure 4: Visual comparisons between real images (left 3 panels) and generated images (right 3 panels) by EPT-LSDR-$\chi^2$ on MNIST, CIFAR10 and CelebA.

## 7 CONCLUSION

EPT is a new approach for generative learning via training a transport map that pushes forward a reference to the target. This approach uses the forward Euler method for solving the McKean-Vlasov equation, which results from linearizing the Monge-Ampère equation that characterizes the optimal transport map. The EPT map is a composition of a sequence of simple residual maps. The key task in training is the estimation of density ratios that completely determine the residual maps. We estimate density ratios based on the Bregman divergence with gradient penalty using deep density-ratio fitting. We establish bounds on the approximation errors due to linearization, discretization, and density-ratio estimation. Our results provide strong theoretical guarantees for the proposed method and ensure that the EPT map converges fast to the target. We also show that the proposed density-ratio (difference) estimators do not suffer from the "curse of dimensionality" if data is supported on a lower-dimensional manifold. This is an interesting result in itself since density-ratio estimation is an important basic problem in machine learning and statistics. Because EPT is easy to train, computationally stable, and

enjoys strong theoretical guarantees, we expect it to be a useful addition to the methods for generating learning.

The proposed EPT method is motivated from the Monge-Ampère equation that characterizes the optimal transport map. However, while the EPT map pushes forward a reference distribution to the target, it is not an estimate of the optimal transport map itself. How to consistently estimate the Monge-Ampére optimal map is a challenging and open problem.

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

**APPENDIX**

In the appendix, we provide the implementation details on numerical settings, network structures, SGD optimizers, and hyper-parameters in the paper. We show the numerical convergence of EPT with simulated datasets and compare the learning and inference of EPT with other generative models. We give detailed theoretical background and proofs of the results mentioned in the paper. We also provide proofs MMD flow and SVGD can be derived from EPT by choosing appropriate $f$-divergences.

# A    APPENDIX: NUMERICAL EXPERIMENTS

## A.1    IMPLEMENTATION DETAILS, NETWORK STRUCTURES, HYPER-PARAMETERS

We provide the details of two versions of the EPT algorithm, EPTv1 in Algorithm 1 and EPTv2 in Algorithm 2 below. In Algorithm 1, we describe the algorithm without outer loops. In Algorithm 2, we describe the algorithm with a latent structure and outer loops.

### A.1.1    2D EXAMPLES

Experiments on 2D examples in our work were performed with deep LSDR fitting and the Pearson $\chi^2$ divergence. We use the EPTv1 (Algorithm 1) without outer loops. In inner loops, only a multilayer perceptron (MLP) was utilized for dynamic estimation of the density ratio between the model distribution $q_k$ and the target distribution $p$. The network structure and hyper-parameters in EPT and deep LSDR fitting were shared in all 2D experiments. We adopt EPT to push particles from a predrawn pool consisting of 50k i.i.d. Gaussian particles to evolve in 20k steps. We used RMSProp with the learning rate 0.0005 and the batch size 1k as the SGD optimizer. The details are given in Table A1 and Table A2. We note that $s$ is the step size, $n$ is the number of particles, $\alpha$ is the penalty coefficient, and $T$ is the mini-batch gradient descent times of deep LSDR fitting or deep logistic regression in each inner loop hereinafter.

Table A1: MLP for deep LSDR fitting.

| Layer | Details | Output size |
|-------|--------------|-------------|
| 1 | Linear, ReLU | 64 |
| 2 | Linear, ReLU | 64 |
| 3 | Linear, ReLU | 64 |
| 4 | Linear | 1 |

Table A2: Hyper-parameters in EPT on 2D examples.

| Parameter | $s$ | $n$ | $\alpha$ | $T$ |
|-----------|-------|-----|----------|-----|
| Value | 0.005 | 50k | 0 or 0.5 | 5 |

### A.1.2    REAL IMAGE DATA

**Datasets.** We evaluated EPT on three benchmark datasets including two small datasets MNIST, CIFAR10 and one large dataset CelebA from GAN literature. MNIST contains a training set of 60k examples and a test set of 10k examples as $28 \times 28$ bilevel images which were resized to $32 \times 32$ resolution. There are a training set of 50k examples and a test set of 10k examples as $32 \times 32$ color images in CIFAR10. We randomly divided the 200k celebrity images in CelebA into two sets for training and test according to the ratio 9:1. We also pre-processed CelebA images by first taking a $160 \times 160$ central crop and then resizing to the $64 \times 64$ resolution. Only the training sets are used to train our models.

---

**Algorithm 1:** EPTv1: Euler particle transport

---

**Input**: $K \in \mathbb{N}^*, s > 0, \alpha > 0$            `// maximum loop count, step size,`
`regularization coeficient`
$X_i \sim \nu, \tilde{Y}_i^0 \sim \mu, i = 1, 2, \cdots, n$            `// real samples, initial particles`
$k \leftarrow 0$
**while** $k < K$ **do**

$\quad\quad \widehat{R}_\phi^k \in \arg\min_{R_\phi} \frac{1}{n} \sum_{i=1}^n [R_\phi(X_i)^2 + \alpha \|\nabla R_\phi(X_i)\|_2^2 - 2 R_\phi(\tilde{Y}_i^k)]$ via SGD
$\quad\quad$ `// determine the density ratio`
$\quad\quad \hat{\boldsymbol{v}}^k(\boldsymbol{x}) = -f''(\widehat{R}_\phi^k(\boldsymbol{x})) \nabla \widehat{R}_\phi^k(\boldsymbol{x})$            `// approximate the velocity field`
$\quad\quad \widehat{\mathcal{T}}^k = \mathbb{1} + s \hat{\boldsymbol{v}}^k$            `// define the forward Euler map`
$\quad\quad \tilde{Y}_i^{k+1} = \widehat{\mathcal{T}}^k(\tilde{Y}_i^k), i = 1, 2, \cdots, n$            `// update particles`
$\quad\quad k \leftarrow k + 1$
**end**
**Output**: $\tilde{Y}_i^k \sim \tilde{\mu}_k, i = 1, 2, \cdots, n$            `// transported particles`

---

**Evaluation metrics.** *Fréchet Inception Distance* (FID) (Heusel et al., 2017) computes the Wasserstein distance $\mathcal{W}_2$ with summary statistics (mean $\mu$ and variance $\Sigma$) of real samples $\mathbf{x}s$ and generated samples $\mathbf{g}s$ in the feature space of the Inception-v3 model (Szegedy et al., 2016), i.e., FID $=$ $\|\mu_{\mathbf{x}} - \mu_{\mathbf{g}}\|_2^2 + \text{Tr}(\Sigma_{\mathbf{x}} + \Sigma_{\mathbf{g}} - 2(\Sigma_{\mathbf{x}}\Sigma_{\mathbf{g}})^{\frac{1}{2}})$. Here, FID is reported with the TensorFlow implementation and lower FID is better.

**Network architectures and hyper-parameter settings.** We employed the ResNet architectures used by Gao et al. (2019) in our EPT algorithm. Especially, the batch normalization (Ioffe & Szegedy, 2015) and the spectral normalization (Miyato et al., 2018) of networks were omitted for EPT-LSDR-$\chi^2$. To train neural networks, we set SGD optimizers as RMSProp with the learning rate 0.0001 and the batch size 100. Inputs $\{Z_i\}_{i=1}^n$ in EPTv2 (Algorithm 2) were vectors generated from a 128-dimensional standard normal distribution on all three datasets. Hyper-parameters are listed in Table A3 where $IL$ expresses the number of inner loops in each outer loop. Even without outer loops, EPTv1 (Algorithm 1) can generate images on MNIST and CIFAR10 as well by making use of a large set of particles. Table A4 shows the hyper-parameters.

Table A3: Hyper-parameters in EPT **with** outer loops on real image datasets.

| Parameter | $\ell$ | $s$ | $n$ | $\alpha$ | $T$ | $IL$ |
|---|---|---|---|---|---|---|
| Value | 128 | 0.5 | 1k | 0 | 1 | 20 |

Table A4: Hyper-parameters in EPT **without** outer loops on real image datasets.

| Parameter | $s$ | $n$ | $\alpha$ | $T$ |
|---|---|---|---|---|
| Value | 0.5 | 4k | 0 | 5 |

## A.2 NUMERICAL CONVERGENCE

We illustrate the convergence property of the learning dynamics of EPTv1 on synthetic datasets *pinwheel, checkerboard* and *2spirals*. As shown in Figure 5, on the three test datasets, the dynamics of both the estimated LSDR fitting losses in (14) with $\alpha = 0$ and the estimated value of the gradient norms $\mathbb{E}_{X \sim q_k}[\|\nabla R_\phi(X)\|_2]$ demonstrate the estimated LSDR loss converges to the theoretical value $-1$.

**Algorithm 2:** EPTv2: Euler particle transport with latent structure

**Input**: $IL, OL \in \mathbb{N}^*, s > 0, \alpha > 0$        `// maximum inner loop count, maximum`
`outer loop count, step size, regularization coeficient`
$X_i \sim \nu, i = 1, 2, \cdots, n$        `// real samples`
$\widehat{G}_\theta^0 \leftarrow G_\theta^{init}$        `// initialize the transport map`
$j \leftarrow 0$
`/* outer loop */`
**while** $j < OL$ **do**
    $Z_i^j \sim \tilde{\mu}, i = 1, 2, \cdots, n$        `// latent particles`
    $\tilde{Y}_i^0 = \widehat{G}_\theta^j(Z_i^j), i = 1, 2, \cdots, n$        `// intermediate particles`
    $k \leftarrow 0$
    `/* inner loop */`
    **while** $k < IL$ **do**
        $\widehat{R}_\phi^k \in \arg\min_{R_\phi} \frac{1}{n} \sum_{i=1}^n [R_\phi(X_i)^2 + \alpha \|\nabla R_\phi(X_i)\|_2^2 - 2R_\phi(\tilde{Y}_i^k)]$ via SGD
        `// determine the density ratio`
        $\hat{\boldsymbol{v}}^k(\boldsymbol{x}) = -f''(\widehat{R}_\phi^k(\boldsymbol{x}))\nabla \widehat{R}_\phi^k(\boldsymbol{x})$        `// approximate the velocity field`
        $\widehat{\mathcal{T}}^k = \mathbb{1} + s\hat{\boldsymbol{v}}^k$        `// define the forward Euler map`
        $\tilde{Y}_i^{k+1} = \widehat{\mathcal{T}}^k(\tilde{Y}_i^k), i = 1, 2, \cdots, n$        `// update particles`
        $k \leftarrow k + 1$
    **end**
    $\widehat{G}_\theta^{j+1} \in \arg\min_{G_\theta} \frac{1}{n} \sum_{i=1}^n \|G_\theta(Z_i^j) - \tilde{Y}_i^{IL}\|_2^2$ via SGD      `// fit the transport`
`map`
    $j \leftarrow j + 1$
**end**
**Output**: $\widehat{G}_\theta^{OL} : \mathbb{R}^\ell \rightarrow \mathbb{R}^d$        `// transport map with latent structure`

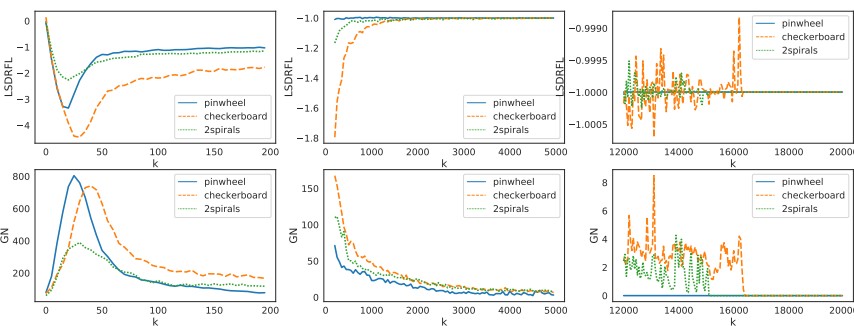

Figure 5: The numerical convergence phenomenon of EPTv1 on simulated datasets. First row: LSDR fitting loss (14) with $\alpha = 0$ v.s. iterations on *pinwheel, checkerboard* and *2spirals*. Second row: Estimation of the gradient norm $\mathbb{E}_{X \sim q_k}[\|\nabla R_\phi(X)\|_2]$ v.s. iterations on *pinwheel, checkerboard* and *2spirals*.

### A.3 LEARNING AND INFERENCE

The learning process of EPT performs particle evolution via solving the McKean-Vlasov equation using forward Euler iterations. The iterations rely on the estimation of the density ratios (difference) between the pushforward distributions and the target distribution. To make the inference of EPTv1 more amendable, we propose EPTv2 based on EPTv1. EPTv2 takes advantage of a neural network to fit the pushforward map. The inference of EPTv2 is fast since the pushforward map is parameterized as a neural network and only forward propagation is involved. These aspects distinguish EPTv2 from score-based generative models Song & Ermon (2019; 2020) which simulate Langevin dynamics to generate samples.

## B  APPENDIX: THEORETICAL BACKGROUNDS AND PROOFS

### B.1  THEORETICAL BACKGROUNDS FOR SECTION 2

For convenience, we first give the following notation to be used in this section. Let $\mathcal{P}_2(\mathbb{R}^m)$ denote the space of Borel probability measures on $\mathbb{R}^m$ with finite second moments, and let $\mathcal{P}_2^a(\mathbb{R}^m)$ denote the subset of $\mathcal{P}_2(\mathbb{R}^m)$ in which measures are absolutely continuous with respect to the Lebesgue measure (all distributions are assumed to satisfy this assumption hereinafter). $\mathrm{Tan}_\mu \mathcal{P}_2(\mathbb{R}^m)$ denotes the tangent space to $\mathcal{P}_2(\mathbb{R}^m)$ at $\mu$. Let $\mathrm{AC}_{\mathrm{loc}}(\mathbb{R}^+, \mathcal{P}_2(\mathbb{R}^m)) := \{\mu_t : I \to \mathcal{P}_2(\mathbb{R}^m)$ is absolutely continuous, $|\mu_t'| \in L^2(I), I \subset \mathbb{R}^+\}$. $\mathrm{Lip}_{\mathrm{loc}}(\mathbb{R}^m)$ denotes the set of functions that are Lipschitz continuous on any compact set of $\mathbb{R}^m$. For any $\ell \in [1, \infty]$, we use $L^\ell(\mu, \mathbb{R}^m)$ ($L^\ell_{\mathrm{loc}}(\mu, \mathbb{R}^m)$) to denote the $L^\ell$ space of $\mu$-measurable functions on $\mathbb{R}^m$ (on any compact set of $\mathbb{R}^m$). With $\mathbb{1}$, $\det$ and $\mathrm{tr}$, we refer to the identity map, the determinant and the trace. We use $\nabla, \nabla^2$ and $\Delta$ to denote the gradient or Jacobian operator, the Hessian operator and the Laplace operator, respectively.

We are now ready to describe the proposed method in a mathematically rigorous fashion and provide theoretical guarantees. Let $X \sim q$, $\widetilde{X} = \mathcal{T}_{t,\Phi}(X)$, and denote the distribution of $\widetilde{X}$ as $\widetilde{q}$. With a small $t$, the map $\mathcal{T}_{t,\phi}$ is invertible according to the implicit function theorem, and we have the change of variables formula

$$\det(\nabla^2 \Psi)(\boldsymbol{x}) = |\det(\nabla \mathcal{T}_{t,\Phi})(\boldsymbol{x})| = \frac{q(\boldsymbol{x})}{\widetilde{q}(\widetilde{\boldsymbol{x}})}, \tag{B-1}$$

where

$$\widetilde{\boldsymbol{x}} = \mathcal{T}_{t,\Phi}(\boldsymbol{x}). \tag{B-2}$$

Using the fact $\frac{\mathrm{d}}{\mathrm{d}t}\big|_{t=0} \det(\mathbf{A} + t\mathbf{B}) = \det(\mathbf{A})\mathrm{tr}\left(\mathbf{A}^{-1}\mathbf{B}\right) \,\forall \mathbf{A}, \mathbf{B} \in \mathbb{R}^{m \times m}$ with $\mathbf{A}$ invertible, and applying the first order Taylor expansion to (B-1), we have

$$\log \widetilde{q}(\widetilde{\boldsymbol{x}}) - \log q(\boldsymbol{x}) = -t\Delta\Phi(\boldsymbol{x}) + o(t). \tag{B-3}$$

Let $t \to 0$ in (B-2) and (B-3), we obtain a random process $\{\boldsymbol{x}_t\}$ and its law $q_t$ satisfying

$$\frac{\mathrm{d}\boldsymbol{x}_t}{\mathrm{d}t} = \nabla\Phi(\boldsymbol{x}_t), \quad \text{with } \boldsymbol{x}_0 \sim q, \tag{B-4}$$

$$\frac{\mathrm{d}\ln q_t(\boldsymbol{x}_t)}{\mathrm{d}t} = -\Delta\Phi(\boldsymbol{x}_t), \quad \text{with } q_0 = q. \tag{B-5}$$

Equations (B-4) and (B-5) resulting from linearization of the Monge-Ampère equation (2) can be interpreted as gradient flows in measure spaces (Ambrosio et al., 2008). And thanks to this connection, we can resort to solving a continuity equation characterized by a type of McKean-Vlasov equation, an ODE system that is easier to handle.

### B.2  GRADIENT FLOWS IN $\mathcal{P}_2^a(\mathbb{R}^m)$

For $\mu \in \mathcal{P}_2^a(\mathbb{R}^m)$ with density $q$, let

$$\mathcal{L}[\mu] = \int_{\mathbb{R}^m} F(q(\boldsymbol{x}))\mathrm{d}\boldsymbol{x} : \mathcal{P}_2^a(\mathbb{R}^m) \to \mathbb{R}^+ \cup \{0\} \tag{B-6}$$

be an energy functional satisfying $\nu \in \arg\min \mathcal{L}[\cdot]$, where $F(\cdot) : \mathbb{R}^+ \to \mathbb{R}^1$ is a twice-differentiable convex function. Among the widely used metrics on $\mathcal{P}_2^a(\mathbb{R}^m)$ in implicit generative learning, the following two are important examples of $\mathcal{L}[\cdot]$ : (1) $f$-divergence given in (5) (Ali & Silvey, 1966); (2) Lebesgue norm of density difference:

$$\|\mu - \nu\|_{L^2(\mathbb{R}^m)}^2 = \int_{\mathbb{R}^m} |q(\boldsymbol{x}) - p(\boldsymbol{x})|^2 \mathrm{d}\boldsymbol{x}. \tag{B-7}$$

**Definition.** We call $\{\mu_t\}_{t\in\mathbb{R}^+} \subset \mathrm{AC}_{\mathrm{loc}}(\mathbb{R}^+, \mathcal{P}_2(\mathbb{R}^m))$ a gradient flow of the functional $\mathcal{L}[\cdot]$, if $\{\mu_t\}_{t\in\mathbb{R}^+} \subset \mathcal{P}_2^a(\mathbb{R}^m)$ a.e., $t \in \mathbb{R}^+$ and the velocity vector field $\boldsymbol{v}_t \in \mathrm{Tan}_{\mu_t}\mathcal{P}_2(\mathbb{R}^m)$ satisfies $\boldsymbol{v}_t \in -\partial\mathcal{L}[\mu_t]$ a.e. $t \in \mathbb{R}^+$, where $\partial\mathcal{L}[\cdot]$ is the subdifferential of $\mathcal{L}[\cdot]$.

The gradient flow $\{\mu_t\}_{t\in\mathbb{R}^+}$ of $\mathcal{L}[\cdot]$ enjoys the following nice properties.

**Proposition B.1.** *(i) The following continuity equation holds in the sense of distributions.*

$$\frac{\partial}{\partial t}\mu_t = -\nabla \cdot (\mu_t \boldsymbol{v}_t) \text{ in } \mathbb{R}^+ \times \mathbb{R}^m \text{ with } \mu_0 = \mu, \tag{B-8}$$

*(ii) Energy decay along the gradient flow: $\frac{\mathrm{d}}{\mathrm{d}t}\mathcal{L}[\mu_t] = -\|\boldsymbol{v}_t\|^2_{L^2(\mu_t, \mathbb{R}^m)}$ a.e. $t \in \mathbb{R}^+$. In addition, $\mathcal{W}_2(\mu_t, \nu) = \mathcal{O}(\exp^{-\lambda t})$, if $\mathcal{L}[\mu]$ is $\lambda$-geodetically convex with $\lambda > 0$ [1].*

*(iii) Conversely, if $\{\mu_t\}_t$ is the solution of continuity equation (B-8) in (i) with $\boldsymbol{v}_t(\boldsymbol{x})$ specified by (B-9) in (ii), then $\{\mu_t\}_t$ is a gradient flow of $\mathcal{L}[\cdot]$.*

**Remark B.1.** *In part (ii) of Proposition B.1, for general $f$-divergences, we assume the functional $\mathcal{L}$ to be $\lambda$-geodesically convex for the convergence of $\mu_t$ to the target $\nu$ in the quadratic Wasserstein distance. However, for the KL divergence, the convergence can be guaranteed if $\nu$ satisfies the log-Sobolev inequality(Otto & Villani, 2000). In addition, the distributions that are strongly log-concave outside a bounded region, but not necessarily log-concave inside the region satisfy the log-Sobolev inequality, see, for example, Holley & Stroock (1987). Here the functional $\mathcal{L}$ can even be nonconvex, an example includes the densities with double-well potential.*

**Remark B.2.** *Equation (8.48) in Proposition 8.4.6 of and Ambrosio et al. (2008) shows the connection (locally) of the velocity $v_t$ of the gradient flow $\mu_t$ and the optimal transport along $\mu_t$, i.e., let $T^{\mu_{t+h}}_{\mu_t}$ be the optimal transport from $\mu_t$ to $\mu_{t+h}$, then $T^{\mu_{t+h}}_{\mu_t} = I + hv_t + o(h)$ in $L^p$. So locally, $I + hv_t$ approximates the optimal transport map from $\mu_t$ to $\mu_{t+h}$ on $[t, t+h]$ for a small $h$.*

*Proof.* (i) The continuity equation (B-8) follows from the definition of the gradient flow directly, see, page 281 in (Ambrosio et al., 2008). (ii) The first equality follows from the chain rule and integration by part, see, Theorem 24.2 of Villani (2008). The second one on linear convergence follows from Theorem 24.7 of Villani (2008), where the assumption on $\lambda$ in equation (24.6) is equivalent to the $\lambda$-geodetically convex assumption here. (iii) Similar to (i) see, page 281 in Ambrosio et al. (2008). □

**Theorem B.1.** *(i) Representation of the velocity fields: if the density $q_t$ of $\mu_t$ is differentiable, then*

$$\boldsymbol{v}_t(\boldsymbol{x}) = -\nabla F'(q_t(\boldsymbol{x})) \ \mu_t\text{-a.e. } \boldsymbol{x} \in \mathbb{R}^m. \tag{B-9}$$

*(ii) If we let $\Phi$ be time-dependent in (B-4)–(B-5), i.e., $\Phi_t$, then the linearized Monge-Ampère equations (B-4)–(B-5) are the same as the continuity equation (B-8) by taking $\Phi_t(\boldsymbol{x}) = -F'(q_t(\boldsymbol{x}))$.*

*Proof.* (i) Recall $\mathcal{L}[\mu]$ is a functional on $\mathcal{P}_2^a(\mathbb{R}^m)$. By the classical results in calculus of variation (Gelfand & Fomin, 2000),

$$\frac{\partial \mathcal{L}[q]}{\partial q}(\boldsymbol{x}) = \frac{\mathrm{d}}{\mathrm{d}t}\mathcal{L}[q + tg] \mid_{t=0} = F'(q(\boldsymbol{x})),$$

where $\frac{\partial \mathcal{L}[q]}{\partial q}$ denotes the first order of variation of $\mathcal{L}[\cdot]$ at $q$, and $q, g$ are the densities of $\mu$ and an arbitrary $\xi \in \mathcal{P}_2^a(\mathbb{R}^m)$, respectively. Let

$$L_F(z) = zF'(z) - F(z) : \mathbb{R}^1 \to \mathbb{R}^1.$$

Some algebra shows,

$$\nabla L_F(q(\boldsymbol{x})) = q(\boldsymbol{x})\nabla F'(q(\boldsymbol{x})).$$

Then, it follows from Theorem 10.4.6 in (Ambrosio et al., 2008) that

$$\nabla F'(q(\boldsymbol{x})) = \partial^o L(\mu),$$

---

[1] We say that $\mathcal{L}$ is $\lambda$-geodetically convex if there exists a constant $\lambda > 0$ such that for every $\mu_1, \mu_2 \in \mathcal{P}_2^a(\mathbb{R}^m)$, there exists a constant speed geodestic $\gamma : [0, 1] \to \mathcal{P}_2^a(\mathbb{R}^m)$ such that $\gamma_0 = \mu_1, \gamma_1 = \mu_2$ and

$$\mathcal{L}(\gamma_s) \leq (1-s)\mathcal{L}(\mu_1) + s\mathcal{L}(\mu_2) - \frac{\lambda}{2}s(1-s)d(\mu_1, \mu_2), \ \forall s \in [0, 1],$$

where $d$ is a metric defined on $\mathcal{P}_2^a(\mathbb{R}^m)$ such as the quadratic Wasserstein distance.

where, $\partial^o L(\mu)$ denotes the one in $\partial L(\mu)$ with minimum length. The above display and the definition of gradient flow implies the representation of the velocity fields $\boldsymbol{v}_t$.

(ii) The time dependent form of (B-4)-(B-5) reads

$$\frac{\mathrm{d}\boldsymbol{x}_t}{\mathrm{d}t} = \nabla\Phi_t(\boldsymbol{x}_t), \;\; \text{with} \;\; \boldsymbol{x}_0 \sim q,$$

$$\frac{\mathrm{d}\ln q_t(\boldsymbol{x}_t)}{\mathrm{d}t} = -\Delta\Phi_t(\boldsymbol{x}_t), \;\; \text{with} \;\; q_0 = q.$$

By chain rule and substituting the first equation into the second one, we have

$$\frac{1}{q_t}\Big(\frac{\mathrm{d}q_t}{\mathrm{d}t} + \frac{\mathrm{d}q_t}{\mathrm{d}\boldsymbol{x}_t}\frac{\mathrm{d}\boldsymbol{x}_t}{\mathrm{d}t}\Big) = \frac{1}{q_t}\Big(\frac{\mathrm{d}q_t}{\mathrm{d}t} + \nabla q_t \nabla\Phi_t(\boldsymbol{x}_t)\Big)$$

$$= -\Delta\Phi_t(\boldsymbol{x}_t),$$

which implies,

$$\frac{\mathrm{d}q_t}{\mathrm{d}t} = -q_t\Delta\Phi_t(\boldsymbol{x}_t) - \nabla q_t\nabla\Phi_t(\boldsymbol{x}_t) = -\nabla\cdot(q_t\nabla\Phi_t).$$

By (B-9), the above display coincides with the continuity equation (B-8) with $\boldsymbol{v}_t = \nabla\Phi_t = -\nabla F'(q_t(\boldsymbol{x}))$. $\qquad\square$

Theorem B.1 and Proposition B.1 imply that $\{\mu_t\}_t$, the solution of the continuity equation (B-8) with $\boldsymbol{v}_t(\boldsymbol{x}) = -\nabla F'(q_t(\boldsymbol{x}))$, converges rapidly to the target distribution $\nu$. Furthermore, the continuity equation has the following representation under mild regularity conditions on the velocity fields.

**Theorem B.2.** *Assume* $\|\boldsymbol{v}_t\|_{L^1(\mu_t,\mathbb{R}^m)} \in L^1_{\mathrm{loc}}(\mathbb{R}^+)$ *and* $\mathrm{v}_t(\cdot) \in \mathrm{Lip}_{\mathrm{loc}}(\mathbb{R}^m)$ *with upper bound* $B_t$ *and Lipschitz constant* $L_t$ *such that* $(B_t + L_t) \in L^1_{\mathrm{loc}}(\mathbb{R}^+)$. *Then the solution of the continuity equation (B-8) can be represented as* $\mu_t = (\mathbf{X}_t)_{\#}\mu$, *where* $\mathbf{X}_t(\boldsymbol{x}) : \mathbb{R}^+ \times \mathbb{R}^m \to \mathbb{R}^m$ *satisfies the McKean-Vlasov equation (4).*

*Proof.* The Lipschitz assumption of $\boldsymbol{v}_t$ implies the existence and uniqueness of the McKean-Vlasov equation (4) according to the classical results in ODE (Arnold, 2012). By the uniqueness of the continuity equation, see Proposition 8.1.7 in Ambrosio et al. (2008), it is sufficient to show that $\mu_t = (\mathbf{X}_t)_{\#}\mu$ satisfies the continuity equation (B-8) in a weak sense. This can be done by the standard test function and smoothing approximation arguments, see, Theorem 4.4 in Santambrogio (2015) for details. $\qquad\square$

As shown in Lemma B.1 below, the velocity fields associated with the $f$-divergence (5) and the Lebesgue norm (B-7) are determined by density ratio and density difference respectively.

**Lemma B.1.** *The velocity fields* $\boldsymbol{v}_t$ *satisfy*

$$\boldsymbol{v}_t(\boldsymbol{x}) = \begin{cases} -f''(r_t(\boldsymbol{x}))\nabla r_t(\boldsymbol{x}), \;\; \mathcal{L}[\mu] = \mathbb{D}_f(\mu\|\nu), \;\text{where}\; r_t(\boldsymbol{x}) = \frac{q_t(\boldsymbol{x})}{p(\boldsymbol{x})}, \\ -2\nabla d_t(\boldsymbol{x}), \;\; \mathcal{L}[\mu] = \|\mu - \nu\|^2_{L^2(\mathbb{R}^m)}, \;\text{where}\; d_t(\boldsymbol{x}) = q_t(\boldsymbol{x}) - p(\boldsymbol{x}). \end{cases}$$

*Proof.* By definition,

$$F(q_t(\boldsymbol{x})) = \begin{cases} p(\boldsymbol{x})f(\frac{q_t(\boldsymbol{x})}{p(\boldsymbol{x})}), \;\; \mathcal{L}[\mu] = \mathbb{D}_f(\mu\|\nu), \\ (q_t(\boldsymbol{x}) - p(\boldsymbol{x}))^2, \;\; \mathcal{L}[\mu] = \|\mu - \nu\|^2_{L^2(\mathbb{R}^m)}. \end{cases}$$

Direct calculation shows

$$F'(q_t(\boldsymbol{x})) = \begin{cases} f'(\frac{q_t(\boldsymbol{x})}{p(\boldsymbol{x})}), \;\; \mathcal{L}[\mu] = \mathbb{D}_f(\mu\|\nu), \\ 2(q_t(\boldsymbol{x}) - p(\boldsymbol{x})), \;\; \mathcal{L}[\mu] = \|\mu - \nu\|^2_{L^2(\mathbb{R}^m)}. \end{cases}$$

Then, the desired result follows from the above display and (B-9). $\qquad\square$

Several methods have been developed to estimate density ratio and density difference in the literature. Examples include probabilistic classification approaches, moment matching and direct density-ratio (difference) fitting, see Sugiyama et al. (2012a;b); Kanamori & Sugiyama (2014); Mohamed & Lakshminarayanan (2016) and the references therein.

**Proposition B.2.** *Suppose that the velocity fields $\boldsymbol{v}_t$ are Lipschitz continuous with respect to $(\boldsymbol{x}, \mu_t)$, that is, there exists a finite constant $L_{\boldsymbol{v}} > 0$ such that*

$$\|\boldsymbol{v}_t(\boldsymbol{x}) - \boldsymbol{v}_{\tilde{t}}(\tilde{\boldsymbol{x}})\| \leq L_{\boldsymbol{v}}[\|\boldsymbol{x} - \tilde{\boldsymbol{x}}\| + \mathcal{W}_2(\mu_t, \mu_{\tilde{t}})], t, \tilde{t} > 0 \text{ and } \boldsymbol{x}, \tilde{\boldsymbol{x}} \in \mathbb{R}^m. \tag{B-10}$$

*Then for any finite $T > 0$, the bound (11) on the discretization error holds:*

$$\sup_{t \in [0,T]} \mathcal{W}_2(\mu_t, \mu_t^s) = \mathcal{O}(s).$$

**Remark B.3.** *If we take $f(x) = (x-1)^2/2$ in Lemma B.1, then the velocity fields $\boldsymbol{v}_t(\boldsymbol{x}) = \nabla \boldsymbol{r}_t(\boldsymbol{x})$, where $\boldsymbol{r}_t(\boldsymbol{x}) = q_t(\boldsymbol{x})/p(\boldsymbol{x})$. In the proof of Theorem B.1, part (ii), it is shown that $q_t$ satisfies $\mathrm{d}q_t/\mathrm{d}t = -\nabla \cdot (q_t \nabla \Phi_t)$. Thus for this simple $f$-divergence function, the verification of the Lipschitz condition (B-10) amounts to verifying that $\nabla \boldsymbol{r}_t(\boldsymbol{x})$ is Lipschitz in the sense of (B-10).*

*Proof.* Without loss of generality let $K = \frac{T}{s} > 1$ be an integer. Recall $\{\mu_t^s \ t \in [ks, (k+1)s)$ is the piecewise constant interpolation between $\mu_k$ and $\mu_{k+1}$ defined as

$$\mu_t^s = (\mathcal{T}_t^{k,s})_{\#}\mu_k,$$

where,

$$\mathcal{T}_t^{k,s} = \mathbb{1} + (t - ks)\boldsymbol{v}_k,$$

$\mu_k$ is defined in (16)-(18) with $\boldsymbol{v}_k = \boldsymbol{v}_{ks}$, i.e., the continuous velocity in (B-9) at time $ks$, $k = 0, .., K-1$, $\mu_0 = \mu$. Under assumption (B-10) we can first show in a way similar to the proof of Lemma 10 in Arbel et al. (2019) that

$$\mathcal{W}_2(\mu_{ks}, \mu_k) = \mathcal{O}(s). \tag{B-11}$$

Let $\Gamma$ be the optimal coupling between $\mu_k$ and $\mu_{ks}$, and $(X, Y) \sim \Gamma$. Let $X_t = \mathcal{T}_t^{k,s}(X)$ and $Y_t$ be the solution of (4) with $\mathbf{X}_0 = Y$ and $t \in [ks, (k+1)s)$. Then

$$X_t \sim \mu_t^s, \ Y_t \sim \mu_t$$

and

$$Y_t = Y + \int_{ks}^t \boldsymbol{v}_{\tilde{t}}(Y_{\tilde{t}})\mathrm{d}\tilde{t}.$$

It follows that

$$
\begin{aligned}
\mathcal{W}_2^2(\mu_t, \mu_{ks}) &\leq \mathbb{E}[\|Y_t - Y\|_2^2] & \text{(B-12)} \\
&= \mathbb{E}[\|\int_{ks}^t \boldsymbol{v}_{\tilde{t}}(Y_{\tilde{t}})\mathrm{d}\tilde{t}\|_2^2] \\
&\leq \mathbb{E}[(\int_{ks}^t \|\boldsymbol{v}_{\tilde{t}}(Y_{\tilde{t}})\|_2 \mathrm{d}\tilde{t})^2] \\
&\leq \mathcal{O}(s^2).
\end{aligned}
$$

where, the first inequality follows from the definition of $\mathcal{W}_2$, and the last equality follows from the the uniform bounded assumption of $\boldsymbol{v}_t$. Similarly,

$$
\begin{aligned}
\mathcal{W}_2^2(\mu_k, \mu_t^s) &\leq \mathbb{E}[\|X - X_t\|_2^2] \\
&= \mathbb{E}[\|(t - ks)\boldsymbol{v}_k(X)\|_2^2] \\
&\leq \mathcal{O}(s^2). & \text{(B-13)}
\end{aligned}
$$

Then,

$$
\begin{aligned}
\mathcal{W}_2(\mu_t, \mu_t^s) &\leq \mathcal{W}_2(\mu_t, \mu_{ks}) + \mathcal{W}_2(\mu_{ks}, \mu_k) + \mathcal{W}_2(\mu_k, \mu_t^s) \\
&\leq \mathcal{O}(s),
\end{aligned}
$$

where the first inequality follows from the triangle inequality, see for example Lemma 5.3 in Santambrogio (2015), and the second one follows from (B-11)-(B-13). $\qquad\square$

## B.3 Derivation and Proofs of the results in Section 4.

### B.3.1 Bregman score for Density ratio/Difference

The separable Bregman score with the base probability measure $p$ to measure the discrepancy between a measurable function $R : \mathbb{R}^m \to \mathbb{R}^1$ and the density ratio $r$ is

$$\mathfrak{B}_{\mathrm{ratio}}(r, R) = \mathbb{E}_{X \sim p}[g'(R(X))(R(X) - r(X)) - g(R(X))]$$
$$= \mathbb{E}_{X \sim p}[g'(R(X))R(X) - g(R(X))] - \mathbb{E}_{X \sim q}[g'(R(X))].$$

It can be verified that $\mathfrak{B}_{\mathrm{ratio}}(r, R) \geq \mathfrak{B}_{\mathrm{ratio}}(r, r)$, where the equality holds iff $R = r$.

For deep density-difference fitting, a neural network $D : \mathbb{R}^m \to \mathbb{R}^1$ is utilized to estimate the density-difference $d(\boldsymbol{x}) = q(\boldsymbol{x}) - p(\boldsymbol{x})$ between a given density $q$ and the target $p$. The separable Bregman score with the base probability measure $w$ to measure the discrepancy between $D$ and $d$ can be derived similarly,

$$\mathfrak{B}_{\mathrm{diff}}(d, D) = \mathbb{E}_{X \sim p}[w(X)g'(D(X))] - \mathbb{E}_{X \sim q}[w(X)g'(D(X))]$$
$$+ \mathbb{E}_{X \sim w}[g'(D(X))D(X) - g(D(X))].$$

Here, we focus on the widely used least-squares density-ratio (LSDR) fitting with $g(c) = (c - 1)^2$ as a working example for estimating the density ratio $r$. The LSDR loss function is

$$\mathfrak{B}_{\mathrm{LSDR}}(r, R) = \mathbb{E}_{X \sim p}[R(X)^2] - 2\mathbb{E}_{X \sim q}[R(X)] + 1.$$

### B.3.2 Gradient Penalty

We consider a noise convolution form of $\mathfrak{B}_{\mathrm{ratio}}(r, R)$ with Gaussian noise $\boldsymbol{\epsilon} \sim \mathcal{N}(\boldsymbol{0}, \alpha \mathbf{I})$,

$$\mathfrak{B}_{\mathrm{ratio}}^{\alpha}(r, R) = \mathbb{E}_{X \sim p}\mathbb{E}_{\epsilon}[g'(R(X + \boldsymbol{\epsilon}))R(X + \boldsymbol{\epsilon}) - g(R(X + \boldsymbol{\epsilon}))] - \mathbb{E}_{X \sim q}\mathbb{E}_{\epsilon}[g'(R(X + \boldsymbol{\epsilon}))].$$

Taylor expansion applied to $R$ gives

$$\mathbb{E}_{\epsilon}[R(\boldsymbol{x} + \boldsymbol{\epsilon})] = R(\boldsymbol{x}) + \frac{\alpha}{2}\Delta R(\boldsymbol{x}) + \mathcal{O}(\alpha^2).$$

Using equations (13)-(17) in Roth et al. (2017), we get

$$\mathfrak{B}_{\mathrm{ratio}}^{\alpha}(r, R) \approx \mathfrak{B}_{\mathrm{ratio}}(r, R) + \frac{\alpha}{2}\mathbb{E}_p[g''(R)\|\nabla R\|_2^2],$$

i.e., $\frac{1}{2}\mathbb{E}_p[g''(R)\|\nabla R\|_2^2]$ serves as a regularizer for deep density-ratio fitting when $g$ is twice differentiable.

### B.3.3 Proofs in Section 4

Below we prove Theorem 4.1 in Section 4.

**Lemma B.2.** *For given densities $p(\boldsymbol{x})$ and $q(\boldsymbol{x})$, let $r(\boldsymbol{x}) = q(\boldsymbol{x})/p(\boldsymbol{x})$ with $\mathcal{C} = \mathbb{E}_{X \sim q}[r(X)] - 1 < \infty$. For any $\alpha \geq 0$, define a nonnegative functional*

$$\mathfrak{B}_{\mathrm{LSDR}}^{\alpha}(R) = \mathfrak{B}_{\mathrm{LSDR}}(r, R) + \alpha\mathbb{E}_p[\|\nabla R\|_2^2] + \mathcal{C}, \ R \ \text{is measurable}.$$

*Then, $r \in \arg\min_R \mathfrak{B}_{\mathrm{LSDR}}^0(R)$. Moreover, $\mathfrak{B}^{\alpha}(R) = 0$ iff $R(\boldsymbol{x}) = r(\boldsymbol{x}) = 1, \ (q, p)$-a.e. $\boldsymbol{x} \in \mathbb{R}^m$.*

*Proof.* By definition, it is easy to check

$$\mathfrak{B}_{\mathrm{LSDR}}^0(R) = \mathfrak{B}_{\mathrm{ratio}}(r, R) - \mathfrak{B}_{\mathrm{ratio}}(r, r),$$

where $\mathfrak{B}_{\mathrm{ratio}}(r, R)$ is the Bregman score with the base probability measure $p$ between $R$ and $r$. Then $r \in \arg\min_{\mathrm{measureable} \ R} \mathfrak{B}_{\mathrm{LSDR}}^0(R)$ follow from the fact $\mathfrak{B}_{\mathrm{ratio}}(r, R) \geq \mathfrak{B}_{\mathrm{ratio}}(r, r)$ and the equality holds iff $R = r$. Since

$$\mathfrak{B}^{\alpha}(R) = \mathfrak{B}_{\mathrm{LSDR}}^0(R) + \alpha\mathbb{E}_p[\|\nabla R\|_2^2] \geq 0,$$

Then,

$$\mathfrak{B}^{\alpha}(R) = 0$$

iff

$$\mathfrak{B}_{\mathrm{LSDR}}^0(R) = 0 \ \text{and} \ \mathbb{E}_p[\|\nabla R\|_2^2] = 0,$$

which is further equivalent to

$$R = r = \text{constant} \ (q, p)\text{-a.e.},$$

and the constant $= 1$ since $r$ is a density ratio. $\qquad\square$

**Theorem B.3.** *Assume* $\text{supp}(r) = \mathfrak{M}_\epsilon$ *and* $r(\boldsymbol{x})$ *is Lipschitz continuous with the bound $B$ and the Lipschitz constant $L$. Suppose the topological parameter of $\mathcal{H}_{\mathcal{D},\mathcal{W},\mathcal{S},\mathcal{B}}$ in (20) with $\alpha = 0$ satisfies* $\mathcal{D} = \mathcal{O}(\log n)$, $\mathcal{W} = \mathcal{O}(n^{\frac{\mathcal{M}}{2(2+\mathcal{M})}}/\log n)$, $\mathcal{S} = \mathcal{O}(n^{\frac{\mathcal{M}-2}{\mathcal{M}+2}}/\log^4 n)$, *and* $\mathcal{B} = 2B$. *Then,*

$$\mathbb{E}_{\{X_i,Y_i\}_1^n}[\|\widehat{R}_\phi - r\|_{L^2(\nu)}^2] \le C(B^2 + cLm\mathcal{M})n^{-2/(2+\mathcal{M})},$$

*where $C$ is a universal constant.*

*Proof.* We use $\mathfrak{B}(R)$ to denote $\mathfrak{B}_{\text{LSDR}}^0 - C$ for simplicity, i.e.,

$$\mathfrak{B}(R) = \mathbb{E}_{X\sim p}[R(X)^2] - 2\mathbb{E}_{X\sim q}[R(X)]. \tag{B-14}$$

Rewrite (20) with $\alpha = 0$ as

$$\widehat{R}_\phi \in \arg\min_{R_\phi \in \mathcal{H}_{\mathcal{D},\mathcal{W},\mathcal{S},\mathcal{B}}} \widehat{\mathfrak{B}}(R_\phi) = \sum_{i=1}^n \frac{1}{n}(R_\phi(X_i)^2 - 2R_\phi(Y_i)). \tag{B-15}$$

By Lemma B.2 and Fermat's rule (Clarke, 1990), we know $\boldsymbol{0} \in \partial\mathfrak{B}(r)$. Then, $\forall R$ direct calculation yields,

$$\|R - r\|_{L^2(\nu)}^2 = \mathfrak{B}(R) - \mathfrak{B}(r) - \langle\partial\mathfrak{B}(r), R - r\rangle = \mathfrak{B}(R) - \mathfrak{B}(r). \tag{B-16}$$

$\forall\bar{R}_\phi \in \mathcal{H}_{\mathcal{D},\mathcal{W},\mathcal{S},\mathcal{B}}$ we have,

$$\begin{aligned}
\|\widehat{R}_\phi - r\|_{L^2(\nu)}^2 &= \mathfrak{B}(\widehat{R}_\phi) - \mathfrak{B}(r) \\
&= \mathfrak{B}(\widehat{R}_\phi) - \widehat{\mathfrak{B}}(\widehat{R}_\phi) + \widehat{\mathfrak{B}}(\widehat{R}_\phi) - \widehat{\mathfrak{B}}(\bar{R}_\phi) \\
&\quad + \widehat{\mathfrak{B}}(\bar{R}_\phi) - \mathfrak{B}(\bar{R}_\phi) + \mathfrak{B}(\bar{R}_\phi) - \mathfrak{B}(r) \\
&\le 2\sup_{R\in\mathcal{H}_{\mathcal{D},\mathcal{W},\mathcal{S},\mathcal{B}}} |\mathfrak{B}(R) - \widehat{\mathfrak{B}}(R)| + \|\bar{R}_\phi - r\|_{L^2(\nu)}^2,
\end{aligned} \tag{B-17}$$

where the inequality uses the definition of $\widehat{R}_\phi$, $\bar{R}_\phi$ and (B-16). We prove the theorem by upper bounding the expected value of the right hand side term in (B-17). To this end, we need the following auxiliary results (B-18)-(B-20).

$$\mathbb{E}_{\{Z_i\}_i^n}[\sup_R |\mathfrak{B}(R) - \widehat{\mathfrak{B}}(R)|] \le 4C_1(2\mathcal{B} + 1)\mathfrak{G}(\mathcal{H}), \tag{B-18}$$

where

$$\mathfrak{G}(\mathcal{H}) = \mathbb{E}_{\{Z_i,\epsilon_i\}_i^n}\left[\sup_{R\in\mathcal{H}_{\mathcal{D},\mathcal{W},\mathcal{S},\mathcal{B}}} |\frac{1}{n}\sum_{i=1}^n \epsilon_i R(Z_i)|\right]$$

is the Gaussian complexity of $\mathcal{H}_{\mathcal{D},\mathcal{W},\mathcal{S},\mathcal{B}}$ (Bartlett & Mendelson, 2002).

**Proof of (B-18).** Let $g(c) = c^2 - c$, $\boldsymbol{z} = (\boldsymbol{x}, \boldsymbol{y}) \in \mathbb{R}^m \times \mathbb{R}^m$,

$$\widetilde{R}(\boldsymbol{z}) = (g \circ R)(\boldsymbol{z}) = R^2(\boldsymbol{x}) - R(\boldsymbol{y}).$$

Denote $Z = (X, Y)$, $Z_i = (X_i, Y_i), i = 1,...,n$ with $X, X_i$ i.i.d. $\sim p$, $Y, Y_i$ i.i.d. $\sim q$. Let $\widetilde{Z}_i$ be an i.i.d. copy of $Z_i$, and $\sigma_i(\epsilon_i)$ be i.i.d. Rademacher random (standard normal) variables that are independent of $Z_i$ and $\widetilde{Z}_i$. Then,

$$\mathfrak{B}(R) = \mathbb{E}_Z[\widetilde{R}(Z)] = \frac{1}{n}\mathbb{E}_{\widetilde{Z}_i}[\widetilde{R}(\widetilde{Z}_i)],$$

and

$$\widehat{\mathfrak{B}}(R) = \frac{1}{n}\sum_{i=1}^n \widetilde{R}(Z_i).$$

Denote

$$\mathfrak{R}(\mathcal{H}) = \frac{1}{n}\mathbb{E}_{\{Z_i,\sigma_i\}_i^n}[\sup_{R\in\mathcal{H}_{\mathcal{D},\mathcal{W},\mathcal{S},\mathcal{B}}} |\sum_{i=1}^n \sigma_i R(Z_i)|]$$

as the Rademacher complexity of $\mathcal{H}_{\mathcal{D},\mathcal{W},\mathcal{S},\mathcal{B}}$ (Bartlett & Mendelson, 2002). Then,

$$
\begin{aligned}
\mathbb{E}_{\{Z_i\}_i^n}[\sup_R |\mathfrak{B}(R) - \widehat{\mathfrak{B}}(R)|] &= \frac{1}{n}\mathbb{E}_{\{Z_i\}_i^n}[\sup_R |\sum_{i=1}^n (\mathbb{E}_{\widetilde{Z}_i}[\widetilde{R}(\widetilde{Z}_i)] - \widetilde{R}(Z_i))|] \\
&\leq \frac{1}{n}\mathbb{E}_{\{Z_i,\widetilde{Z}_i\}_i^n}[\sup_R |\widetilde{R}(\widetilde{Z}_i) - \widetilde{R}(Z_i)|] \\
&= \frac{1}{n}\mathbb{E}_{\{Z_i,\widetilde{Z}_i,\sigma_i\}_i^n}[\sup_R |\sum_{i=1}^n \sigma_i(\widetilde{R}(\widetilde{Z}_i) - \widetilde{R}(Z_i))|] \\
&\leq \frac{1}{n}\mathbb{E}_{\{Z_i,\sigma_i\}_i^n}[\sup_R |\sum_{i=1}^n \sigma_i\widetilde{R}(Z_i)|] + \frac{1}{n}\mathbb{E}_{\{\widetilde{Z}_i,\sigma_i\}_i^n}[\sup_R |\sum_{i=1}^n \sigma_i\widetilde{R}(\widetilde{Z}_i)|] \\
&= 2\mathfrak{R}(g \circ \mathcal{H}) \\
&\leq 4(2\mathcal{B}+1)\mathfrak{R}(\mathcal{H}) \\
&\leq 4C_1(2\mathcal{B}+1)\mathfrak{G}(\mathcal{H}),
\end{aligned}
$$

where, the first inequality follows from the Jensen's inequality, and the second equality holds since the distribution of $\sigma_i(\widetilde{R}(\widetilde{Z}_i) - \widetilde{R}(Z_i))$ and $\widetilde{R}(\widetilde{Z}_i) - \widetilde{R}(Z_i)$ are the same, and the last equality holds since the distribution of the two terms are the same, and last two inequality follows from the Lipschitz contraction property where the Lipschitz constant of $g$ on $\mathcal{H}_{\mathcal{D},\mathcal{W},\mathcal{S},\mathcal{B}}$ is bounded by $2\mathcal{B}+1$ and the relationship between the Gaussian complexity and the Rademacher complexity, see for Theorem 12 and Lemma 4 in Bartlett & Mendelson (2002), respectively.

$$
\mathfrak{G}(\mathcal{H}) \leq C_2\mathcal{B}\sqrt{\frac{n}{\mathcal{D}\mathcal{S}\log\mathcal{S}}}\log\frac{n}{\mathcal{D}\mathcal{S}\log\mathcal{S}}\exp(-\log^2\frac{n}{\mathcal{D}\mathcal{S}\log\mathcal{S}}). \tag{B-19}
$$

**Proof of (B-19).**
Since $\mathcal{H}$ is negation closed,

$$
\begin{aligned}
\mathfrak{G}(\mathcal{H}) &= \mathbb{E}_{\{Z_i,\epsilon_i\}_i^n}[\sup_{R\in\mathcal{H}_{\mathcal{D},\mathcal{W},\mathcal{S},\mathcal{B}}}\frac{1}{n}\sum_{i=1}^n \epsilon_i R(Z_i)] \\
&= \mathbb{E}_{Z_i}[\mathbb{E}_{\epsilon_i}[\sup_{R\in\mathcal{H}_{\mathcal{D},\mathcal{W},\mathcal{S},\mathcal{B}}}\frac{1}{n}\sum_{i=1}^n \epsilon_i R(Z_i)]|\{Z_i\}_{i=1}^n].
\end{aligned}
$$

Conditioning on $\{Z_i\}_{i=1}^n$, $\forall R, \widetilde{R} \in \mathcal{H}_{\mathcal{D},\mathcal{W},\mathcal{S},\mathcal{B}}$ it easy to check

$$
\mathbb{V}_{\epsilon_i}[\frac{1}{n}\sum_{i=1}^n \epsilon_i(R(Z_i) - \widetilde{R}(Z_i))] = \frac{d_2^{\mathcal{H}}(R,\tilde{R})}{\sqrt{n}},
$$

where, $d_2^{\mathcal{H}}(R,\tilde{R}) = \frac{1}{\sqrt{n}}\sqrt{\sum_{i=1}^n (R(Z_i) - \tilde{R}(Z_i))^2}$. Observing the diameter of $\mathcal{H}_{\mathcal{D},\mathcal{W},\mathcal{S},\mathcal{B}}$ under $d_2^{\mathcal{H}}$ is at most $\mathcal{B}$, we have

$$
\begin{aligned}
\mathfrak{G}(\mathcal{H}) &\leq \frac{C_3}{\sqrt{n}}\mathbb{E}_{\{Z_i\}_{i=1}^n}[\int_0^{\mathcal{B}}\sqrt{\log\mathcal{N}(\mathcal{H},d_2^{\mathcal{H}},\delta)}d\delta] \\
&\leq \frac{C_3}{\sqrt{n}}\mathbb{E}_{\{Z_i\}_{i=1}^n}[\int_0^{\mathcal{B}}\sqrt{\log\mathcal{N}(\mathcal{H},d_\infty^{\mathcal{H}},\delta)}d\delta] \\
&\leq \frac{C_3}{\sqrt{n}}\int_0^{\mathcal{B}}\sqrt{VC_{\mathcal{H}}\log\frac{6\mathcal{B}n}{\delta VC_{\mathcal{H}}}}d\delta, \\
&\leq C_4\mathcal{B}(\frac{n}{VC_{\mathcal{H}}})^{1/2}\log(\frac{n}{VC_{\mathcal{H}}})\exp(-\log^2(\frac{n}{VC_{\mathcal{H}}})) \\
&\leq C_2\mathcal{B}\sqrt{\frac{n}{\mathcal{D}\mathcal{S}\log\mathcal{S}}}\log\frac{n}{\mathcal{D}\mathcal{S}\log\mathcal{S}}\exp(-\log^2\frac{n}{\mathcal{D}\mathcal{S}\log\mathcal{S}})
\end{aligned}
$$

where, the first inequality follows from the chaining Theorem 8.1.3 in Vershynin (2018), and the second inequality holds due to $d_2^{\mathcal{H}} \leq d_\infty^{\mathcal{H}}$, and in the third inequality we used the relationship between

the matric entropy and the VC-dimension of the ReLU networks $\mathcal{H}_{\mathcal{D},\mathcal{W},\mathcal{S},\mathcal{B}}$ (Anthony & Bartlett, 2009), i.e.,

$$\log \mathcal{N}(\mathcal{H}, d_\infty^{\mathcal{H}}, \delta) \leq \mathrm{VC}_\mathcal{H} \log \frac{6\mathcal{B}n}{\delta \mathrm{VC}_\mathcal{H}},$$

and the fourth inequality follows by some calculation, and the last inequality holds due to the upper bound of VC-dimension for the ReLU network $\mathcal{H}_{\mathcal{D},\mathcal{W},\mathcal{S},\mathcal{B}}$ satisfying

$$\mathrm{VC}_\mathcal{H} \leq C_5 \mathcal{D}\mathcal{S} \log \mathcal{S},$$

see Bartlett et al. (2019).
For any two integer $M, N$, there exists a $\bar{R}_\phi \in \mathcal{H}_{\mathcal{D},\mathcal{W},\mathcal{S},\mathcal{B}}$ with width $\mathcal{W} = \max\{8\mathcal{M}N^{1/\mathcal{M}} + 4\mathcal{M}, 12N + 14\}$ and depth $\mathcal{D} = 9M + 12$, and $\mathcal{B} = 2B$, such that

$$\|r - \bar{R}_\phi\|_{L^2(\nu)}^2 \leq C_6 cLm\mathcal{M}(NM)^{-4/\mathcal{M}}. \tag{B-20}$$

**Proof of (B-20).**
We use Lemma 4.1, Theorem 4.3, 4.4 and following the proof of Theorem 1.3 in Shen et al. (2019). Let $\mathbf{A}$ be the random orthoprojector in Theorem 4.4, then it is to check $\mathbf{A}(\mathfrak{M}_\epsilon) \subset \mathbf{A}([-c,c]^m) \subset [-c\sqrt{m}, \sqrt{m}c]^{\mathcal{M}}$. Let $\tilde{r}$ be an extension of the restriction of $r$ on $\mathfrak{M}_\epsilon$, which is defined similarly as $\tilde{g}$ on page 30 in Shen et al. (2019). Since we assume the target $r$ is Lipschitz continuous with the bound $B$ and the Lipschitz constant $L$, let $\epsilon$ small enough, then by Theorem 4.3, there exist a ReLU network $\tilde{R}_\phi \in \mathcal{H}_{\mathcal{D},\mathcal{W},\mathcal{S},\mathcal{B}}$ with width

$$\mathcal{W} = \max\{8\mathcal{M}N^{1/\mathcal{M}} + 4\mathcal{M}, 12N + 14\},$$

and depth

$$\mathcal{D} = 9M + 12,$$

and $\mathcal{B} = 2B$, such that

$$\|\tilde{r} - \tilde{R}_\phi\|_{L^\infty(\mathfrak{M}_\epsilon \setminus \mathcal{N})} \leq 80cL\sqrt{m\mathcal{M}}(NM)^{-2/m},$$

and

$$\|\tilde{R}_\phi\|_{L^\infty(\mathfrak{M}_\epsilon)} \leq B + 3Lc\sqrt{m\mathcal{M}},$$

where, $\mathcal{N}$ is a $\nu-$ negligible set with $\nu(\mathcal{N})$ can be arbitrary small. Define $\bar{R}_\phi = \tilde{R}_\phi \circ \mathbf{A}$. Then, following the proof after equation (4.8) in Theorem 1.3 of Shen et al. (2019), we get our (B-20) and

$$\|\bar{R}_\phi\|_{L^\infty(\mathfrak{M}_\epsilon \setminus \mathcal{N})} \leq 2B, \|\bar{R}_\phi\|_{L^\infty(\mathcal{N})} \leq 2B + 3cL\sqrt{m\mathcal{M}}.$$

Let $\mathcal{D}\mathcal{S} \log \mathcal{S} < n$, combing the results (B-17) - (B-20), we have

$$\mathbb{E}_{\{X_i, Y_i\}_1^n}[\|\widehat{R}_\phi - r\|_{L^2(\nu)}^2]$$
$$\leq 8C_1(2B+1)\mathfrak{G}(\mathcal{H}) + C_6 cLm\mathcal{M}(NM)^{-4/\mathcal{M}}$$
$$\leq 8C_1(2B+1)C_2 B \sqrt{\frac{\mathcal{D}\mathcal{S} \log \mathcal{S}}{n}} \log \frac{n}{\mathcal{D}\mathcal{S} \log \mathcal{S}}$$
$$+ C_6 cLm\mathcal{M}(NM)^{-4/\mathcal{M}}$$
$$\leq C(B^2 + cLm\mathcal{M})n^{-2/(2+\mathcal{M})},$$

where, last inequality holds since we choose $M = \log n$, $N = n^{\frac{\mathcal{M}}{2(2+\mathcal{M})}}/\log n$, $\mathcal{S} = n^{\frac{\mathcal{M}-2}{\mathcal{M}+2}}/\log^4 n$, i.e., $\mathcal{D} = 9 \log n + 12$, $\mathcal{W} = 12n^{\frac{\mathcal{M}}{2(2+\mathcal{M})}}/\log n + 14$. $\qquad\square$

### B.4 THE RELATIONSHIP BETWEEN EPT AND MMD FLOW

Here we show that MMD flow can be considered a special case of EPT.

*Proof.* Let $\mathcal{H}$ be a reproducing kernel Hilbert space with characteristic kernel $K(\boldsymbol{x}, \boldsymbol{z})$. Recall in MMD flow,

$$\mathcal{L}[\mu] = \frac{1}{2}\|\mu - \nu\|_{\mathrm{mmd}}^2,$$

and

$$\frac{\partial \mathcal{L}[\mu]}{\partial \mu}(\boldsymbol{x}) = \int K(\boldsymbol{x}, \boldsymbol{z}) \mathrm{d}\mu(\boldsymbol{z}) - \int K(\boldsymbol{x}, \boldsymbol{z}) \mathrm{d}\nu(\boldsymbol{z}),$$

and the vector fields

$$\begin{aligned}
\boldsymbol{v}_t^{\mathrm{mmd}} &= -\nabla \frac{\partial \mathcal{L}[\mu]}{\partial \mu_t} \\
&= \int \nabla_{\boldsymbol{x}} K(\boldsymbol{x}, \boldsymbol{z}) \mathrm{d}\nu(\boldsymbol{z}) - \int \nabla_{\boldsymbol{x}} K(\boldsymbol{x}, \boldsymbol{z}) \mathrm{d}\mu_t(\boldsymbol{z}) \\
&= \int \nabla_{\boldsymbol{x}} K(\boldsymbol{x}, \boldsymbol{z}) p(\boldsymbol{z}) \mathrm{d}\boldsymbol{z} - \int \nabla_{\boldsymbol{x}} K(\boldsymbol{x}, \boldsymbol{z}) q_t(\boldsymbol{z}) \mathrm{d}\boldsymbol{z}
\end{aligned}$$

By Lemma B.1, the vector fields corresponding the Lebesgue norm $\frac{1}{2}\|\mu - \nu\|_{L^2(\mathbb{R}^m)}^2 = \frac{1}{2}\int_{\mathbb{R}^m} |q(\boldsymbol{x}) - p(\boldsymbol{x})|^2 \mathrm{d}\boldsymbol{x}$ are defined as

$$\boldsymbol{v}_t = \nabla p(\boldsymbol{x}) - \nabla q_t(\boldsymbol{x}).$$

Next, we will show the vector fields $\boldsymbol{v}_t^{\mathrm{mmd}}$ is exactly by projecting the vector fields $\boldsymbol{v}_t$ on to the reproducing kernel Hilbert space $\mathcal{H}^m = \mathcal{H}^{\otimes m}$. By the definition of reproducing kernel we have,

$$p(\boldsymbol{x}) = \langle p(\cdot), K(\boldsymbol{x}, \cdot) \rangle_{\mathcal{H}} = \int K(\boldsymbol{x}, \boldsymbol{z}) p(\boldsymbol{z}) \mathrm{d}\boldsymbol{z},$$

and

$$q_t(\boldsymbol{x}) = \langle q_t(\cdot), K(\boldsymbol{x}, \cdot) \rangle_{\mathcal{H}} = \int K(\boldsymbol{x}, \boldsymbol{z}) q_t(\boldsymbol{z}) \mathrm{d}\boldsymbol{z}.$$

Hence,

$$\begin{aligned}
\boldsymbol{v}_t(\boldsymbol{x}) &= \nabla p(\boldsymbol{x}) - \nabla q_t(\boldsymbol{x}) \\
&= \int \nabla_{\boldsymbol{x}} K(\boldsymbol{x}, \boldsymbol{z})(p(\boldsymbol{z}) - q_t(\boldsymbol{z})) \mathrm{d}\boldsymbol{z} \\
&= \boldsymbol{v}_t^{\mathrm{mmd}}(\boldsymbol{x}).
\end{aligned}$$

This completes the proof. $\qquad\square$

## B.5 Proof of the relation between EPT and SVGD

*Proof.* Let $f(u) = u \log u$ in (5). With this $f$ the velocity fields $\boldsymbol{v}_t = -f''(r_t)\nabla r_t = -\frac{\nabla r_t(\mathbf{x})}{r_t(\mathbf{x})}$ Let $\mathbf{g}$ in a Stein class associated with $q_t$.

$$\begin{aligned}
&\langle \boldsymbol{v}_t, \mathbf{g} \rangle_{\mathcal{H}(q_t)} \\
&= -\int \mathbf{g}(\mathbf{x})^T \frac{\nabla r_t(\mathbf{x})}{r_t(\mathbf{x})} q_t(\mathbf{x}) \mathrm{d}\mathbf{x} \\
&= -\int \mathbf{g}(\mathbf{x})^T \nabla \log r_t(\mathbf{x}) q_t(\mathbf{x}) \mathrm{d}\mathbf{x} \\
&= -\mathbb{E}_{\mathbf{X} \sim q_t(\mathbf{x})}[\mathbf{g}(\mathbf{x})^T \nabla \log q_t(\mathbf{X}) + \mathbf{g}(\mathbf{x})^T \nabla \log p(\mathbf{X})] \\
&= -\mathbb{E}_{\mathbf{X} \sim q_t(\mathbf{x})}[\mathbf{g}(\mathbf{x})^T \nabla \log q_t(\mathbf{X}) + \nabla \cdot \mathbf{g}(\mathbf{x})] \\
&\quad + \mathbb{E}_{\mathbf{X} \sim q_t(\mathbf{x})}[\mathbf{g}(\mathbf{x})^T \nabla \log p(\mathbf{X}) + \nabla \cdot \mathbf{g}(\mathbf{x})] \\
&= -\mathbb{E}_{\mathbf{X} \sim q_t(\mathbf{x})}[\mathcal{T}_{q_t}\mathbf{g}] + \mathbb{E}_{\mathbf{X} \sim q_t(\mathbf{x})}[\mathcal{T}_p\mathbf{g}] \\
&= \mathbb{E}_{\mathbf{X} \sim q_t(\mathbf{x})}[\mathcal{T}_p\mathbf{g}],
\end{aligned}$$

where the last equality is obtained by restricting $\mathbf{g}$ in a Stein class associated with $q_t$, i.e., $\mathbb{E}_{\mathbf{X} \sim q_t(\mathbf{x})}\mathcal{T}_{q_t}\mathbf{g} = 0$. This is exactly the velocity fields of SVGD (Liu, 2017). $\qquad\square$

