# OpenReview forum: "Generative Learning With Euler Particle Transport"
_ICLR.cc/2021/Conference — Reject_

### Official Review · AnonReviewer2 · 2020-10-28
**Interesting study but more rigor needed**

**Rating:** 5
**Confidence:** 4

**Review:**

This paper considers generative learning by discretizing a Wasserstein gradient with Euler methods. More precisely, some samples of a target distribution are given and the goal is to pushforward some samples of an initial distribution to the target distribution. The proposed method is obtained by  minimizing the f-divergence between the initial distribution and the target distribution, but considering the Wasserstein gradient flow of the f-divergence w.r.t. the target distribution (= the objective function). This Wasserstein gradient flow is discretized via Euler method to obtain the proposed algorithm. This Euler method involves the Wasserstein gradient of the objective, which is intractable. The authors describe a statistical methodology to compute this Wasserstein gradient based on the samples of the target distribution and samples from the current distribution. They also prove a bound for the estimated Wasserstein gradient wrt the true Wasserstein gradient. Finally, the paper presents relevant numerical experiments.

Overall the approach is natural, the topic is interesting and paper is well written. The authors show how to approximate the Wasserstein gradient, which is the main technical point in the implementation of the ETP for f-divergences. They also prove a bound for it and the simulations are promising.
However, some typos/approximative statements make the reading experience difficult (see below).
Eq (10). The linear convergence of the gradient flow to the target distribution is not something standard, since it usually requires the objection to be lambda-geo-convex (lambda > 0). It holds if the f-divergence is KL and the target strongly log concave but beyond this case I don't know. Could the author give other examples? Moreover, I don't believe that it is a good strategy to claim strong results like this without justifying.
Eq (11). I don't see why Eq B-10 holds here and I suspect that it does not. Can the authors prove that v_t is jointly Lipschitz? Otherwise, remove Eq (11). Moreover, Eq (11) does not imply convergence to the target distribution as claimed. The reason is that Eq (11) is a uniform convergence result over compact sets [0,T]. This does not imply convergence in the limit T = infty in general, as well documented in many works, see e.g. Kushner & Yin 2003. The situation is much more complicated (an additional stability result would be needed).
Th 4.1. How reasonable is it to assume r Lipschitz. In the EPT, r depends on the current density, whose regularity is unknown. Moreover, How could be combine this bound in a study of the EPT itself. Can we study the convergence of the EPT? I understand that these questions are difficult and might not be answered by the authors now, but they should at least be raised as limitations of the paper. In the current form, the reader has the feeling that the authors just overlooked most of the fundamental questions.
Finally, why the composition of transport maps constructed by the algorithm should approximately be the optimal one? This would depend on the objective function which is an f-divergence. I don't understand the linearization of Monge Ampere equation.

Overall, the paper and the result are interesting but more rigor should be put on justifying the claims. Unjustified claims are not  unnoticed. Moreover, the fundamental questions should be raised. I rate the paper as marginally above, as I appreciate the bound on the Wasserstein gradient which is a first step toward analyzing the ETP (and is relevant from a statistical estimation point of view). The num exp also show interesting results for generative learning tasks.

MINOR :

ETP vs EPT throughout the paper
Section 2: "Lebesque"
Section 3, line 6-7. typos (parentheses). Moreover, some notations are not yet defined
More details (like an intuitive explanation of Lemma 4.1) on the LSDR methodology would be appreciated in the main paper.
After Lemma 4.1, "Lipsichiz"
Th 4.1: "with the bound B" ??
Related works on discretized gradient flows : https://arxiv.org/pdf/2002.03035.pdf, https://arxiv.org/pdf/1704.07520.pdf, https://arxiv.org/abs/2006.02509, https://arxiv.org/abs/2006.09797 among others.
Check end of first paragraph of Page 7.

---

> ### Author Response · Authors · 2020-11-16
> **Response to AnonReviewer 2**
>
> We are grateful to you for taking the time to read our paper and for providing detailed and helpful comments.
> Our responses to your comments are given below.
>
> 1.	On the linear convergence of the gradient flow, i.e., equation (10): Yes, for general f-divergence, the linear convergence holds when we assume the functional is $\lambda$-geodesically convex. But for the KL divergence, this linear convergence can be guaranteed if the target distribution satisfies the log-Sobolev inequality, see Otto and Villani (2000). Most importantly, in addition to the strongly log-concave ones, the distributions that are strongly log-concave outside a bounded region, but not necessarily log-concave inside the region can also satisfy the log-Sobolev inequality, see, for example, Holley and Stroock (1987).  Here the functional can even be nonconvex, an example includes the densities with double-well potential.
>
> 2.	On equation (11): Yes, equation (11) and equation (B-10) depend on the assumption that $v_t$ is jointly Lipschitz continuous. Indeed, this joint Lipschitz condition is not easy to check. This is an interesting and challenging problem. We will keep working on this question.  We agree that equation (11) alone cannot imply convergence as $T$ goes to infinity. However, we do not need this to hold for the following reasons. Equation (10) says that $\mu_t$ converges to the target fast. Therefore, at a large time T, $\mu_T$ is close to the target distribution. So we just need to estimate $\mu_T$ and approximate the target by $\mu_T$. Equation (11) bounds the discretization error in this finite interval $(0,T]$. Hence, equation (10) and (11) together implies that particles defined in (7) with $s$ small and $k$ sufficiently large approximates the target well. We will add this point to the revision.
>
> 3.  On the Lipschitz assumption on $r$: The Lipschitz continuity is a commonly used assumption in non-parametric density estimation. Indeed,  it is indeed difficult to check this condition. Again, this is an interesting and challenging question, we will consider this further.
>
> 4.	On the overall error of EPT: This is a very good question and we have been working on this since we started this work. At the sample level, we can bound the error of the density-ratio estimator, as shown in Theorem 4.1.  However, what we needed is to bound the error of the estimated gradient. This is the main difficulty that prevented us to obtain an overall error bound.
>
> MINOR :
>
> ETP vs EPT throughout the paper Section 2: "Lebesque" Section 3, line 6-7. typos (parentheses).
>
> We will correct these typos.
>
> Moreover, some notations are not yet defined
>
> We will make sure the notations are defined before they are used in the revision.
>
> More details (like an intuitive explanation of Lemma 4.1) on the LSDR methodology would be appreciated in the main paper.
>
> We will add a more detailed explanation of Lemma 4.1 and LSDR in the revision.
>
> After Lemma 4.1, "Lipsichiz"
>
> We will change "Lipsichiz’’ to Lipschitz.
>
> Th 4.1: "with the bound B" ??
>
> We meant r(x) is bounded by a finite constant B. We will rewrite this sentence to make it clear in the revision.
>
> Related works on discretized gradient flows : https://arxiv.org/pdf/2002.03035.pdf, https://arxiv.org/pdf/1704.07520.pdf, https://arxiv.org/abs/2006.02509, https://arxiv.org/abs/2006.09797 among others.
>
> Thank you for pointing out these related references to us. We will have a good read of them.
>
>  Check end of first paragraph of Page 7.
>
> The link is not available due to the anonymous requirement.  We will make it available at a later time.

---

> > ### Comment · AnonReviewer2 · 2020-11-23
> > **Thanks**
> >
> > Thank you for your answers. I notice that the paper has not been updated yet.
> >
> > 0. I also see that you addressed my concern w.r.t optimal pushforward map in the answer to R3.
> > 1. OK for Log Sobolev Inequality.
> > 2. and 3. Indeed these assumptions are not easy to check and might not hold. That's why Forward Euler schemes are difficult to analyze. Perhaps this is the reason why this scheme is not well studied in the literature. I encourage the authors to state the Lipschitz assumption in Prop B2.
> > 2. OK but the O(s) hides a dependence in T which is exponential (see Arbel et al. 19) so s must be very small i.e. many iterations are needed. Still this does not imply convergence.
> >
> > Overall, I am a bit disappointed by the answers so I am lowering my score.

---

> > > ### Author Response · Authors · 2020-11-23
> > > **Response to AnonReviewer2 thank you for your additional comments**
> > >
> > > Thank you for your additional comments. We will submit the revision momentarily.
> > >
> > > 0. We meant to respond to your concern about the optimal transport map. I (as the corresponding author) apologize for inadvertently forgetting to include it in our earlier response.
> > > 1. Thanks.
> > > 2. and 3. Again, we agree that these assumptions are not easy to check. However, this does not mean that they always don't hold. We have stated the Lipschitz condition in Proposition B2 in the revision. We believe that our work is the first to use the forward Euler method in the context of generative learning. Unlike the existing methods on this topic, we not only developed the proposed method but also provided convergence analysis for our method.
> > > 3. Indeed, it is well known that the Euler method needs a small step size s.  This is generally true for any first-order method for solving nonlinear equations. However,  based on our numerical experiments, it does not have to `be "very small."  Specifically, we used s=0.5 in our experiments, see Tables A.3 and A.4 in the appendix. In addition, our numerical experiments confirm that the algorithm based on the proposed method converges fast and is computationally stable.
> > >
> > > We are very disappointed that you are lowering your score. We feel that we got unfairly punished for trying to do error analysis on a very difficult problem that has not been done before. Although the conditions are strong, they do not appear unreasonable under certain conditions on the underlying target distribution. We believe our analysis is a valuable contribution to the literature on this topic,  at least it provides a starting point for further theoretical analysis. In any case, we thank you for reading our paper and for your comments.

---

### Official Review · AnonReviewer3 · 2020-10-30
**The work is technically solid, but may not be doing what is claimed.**

**Rating:** 5
**Confidence:** 4

**Review:**

Pros:
* The exploration to use a flow/transport map for generative modeling is inspiring and worth encouraging.
* The writing roughly follows a clear logic flow.
* The proofs seem valid to me.

Cons:
* Method.
  - My major concern is on whether the method indeed implements the optimal transport map. As I understand it, optimal transport map and gradient flow are two independent concepts. The former is determined by two distributions, while the latter by a function(al) of distribution which gives the gradient. I do not understand why various gradient flows can all approximate the optimal transport map. Particularly, the optimal transport map does not depend on the choice of the function f in the f-divergence, while different choices of f give different gradient flows.

    It makes sense to approach to the target distribution $\nu$ by minimizing the f-divergence of the current distribution $\mu_t$ to $\nu$, which can be done by approximating the gradient flow of the f-divergence. But it does not seem to have a relation to optimal transport, which is the claimed motivation of the work.
* Writing.
  - Some theoretical results seem to be already well-established, e.g., results in Theorem B.1 are all covered by Villani (2008) and Ambrosio et al. (2008). The authors may need to rephrase these results as an introduction to the background knowledge of this area, so that the novel theoretical contributions can be highlighted.
  - The main paper frequently refers to the appendix, which feels a little disturbing.
  - It seems to me that $q(x)$ in Eq. (2) should be $q(z)$, since $q$ is the density of $\mu$, which represents the easy-to-sample measure/distribution on the latent space, and $Z \sim \mu$.

=== EDIT: post-rebuttal ===

I thank the authors for their patient and detailed replies regarding my concerns. I noticed that the authors have addressed the concerns to some extent in the updated paper (e.g., the remark in Conclusion). I'm also aware that the method can achieve the machine-learning goal of transporting a reference distribution to the data distribution, regardless of my concern. So I raised my score by 1 point.

Nevertheless, I still feel uncomfortable to give a positive score. The current presentation of the motivation may confuse or even mislead the community. The authors present the Monge-Ampere Eq. (2), which solves for the _optimal_ transport from the reference distribution to the data distribution. But the method is constructed by simulating the gradient flow of f-divergence. Although the resulting transport serves for transforming the reference distribution to the data distribution (since the gradient flow minimizes their f-divergence), the transport is unnecessarily _optimal_ (there may be multiple transports to transform the reference distribution to the data distribution; the gradient-flow transport is one of them, which may not be the optimal one that solves the Monge-Ampere equation).

For the authors' reply "Our method is based on a first-order approximation to the Monge-Ampere equation", I think the "Monge-Ampere equation" therein is different from the original one (Eq. (2)). If the method, which is constructed from simulating the gradient flow of the f-divergence, is to be treated as a first-order approximation to some Monge-Ampere equation, then the Monge-Ampere equation is between two adjacent/neighboring distributions on the gradient flow curve, but Eq. (2) is between the two ending points of the curve (i.e., the reference distribution and the data distribution). The two Monge-Ampere equations have different solutions unless the gradient flow coincides with the geodesic on the Wasserstein space, which is unnecessarily the case.

I agree that "An important advantage of the proposed approach is that it allows general energy functional in constructing the gradient flow from the reference distribution to the target distribution". But I think it does not have much to do with _optimal_ transport or the Monge-Ampere equation. This thought works since the gradient flow minimizes the energy functional, whose minimum is achieved only when the two distributions coincide (when the energy functional is a proper divergence/discrepancy).

---

> ### Author Response · Authors · 2020-11-16
> **Response to AnonReviewer3**
>
> We would like to thank you for taking the time to read our paper. However, based on the comments we are afraid you may have misunderstood our work and did not provide a fair assessment of our contribution. We hope our responses and the comments by the other reviewers may help clarify the key points of our work and its contribution.
>
> Our proposed method seeks to find an approximation to the optimal transport map. We have never claimed the generative map learned via EPT is the exact optimal transport map. As we highlighted at the beginning of Section 2 and after Eq. (2), the optimal transport map is characterized by the Monge-Ampere equation. However, it is difficult to solve this equation to obtain the exact optimal transport map due to its highly nonlinear nature. So, we propose to use a linear approximation method using residual maps. This linearization approach for studying the Monge-Ampere equation has been developed by many researchers, see, for example, the books by Villani, and Ambrosio et al. The linearization approach leads to the McKean-Vlasov equation, which can be equivalently characterized by gradient flows. This gradient flows with different choices of f lead to different approximations to the optimal transport map. Indeed, a different f generates a different sequence of residuals maps, their composition converges to an approximation of the optimal map. This is proved in Theorem B.1.  The gradient flow will provide a map that will push forward the reference distribution to the target distribution. This map is an approximation to the optimal one, it can be estimated via samples in a computationally feasible way for generative learning.  This is the basic idea behind our method.  We have described this idea clearly in Section 2.
>
> As explained above and in the paper, the optimal transport map is the solution to the Monge-Ampere equation. Because this equation is highly nonlinear and infeasible to solve directly, we use the linear approximation method to obtain a sequence of residual maps determined by the velocity fields associated with the McKean-Vlasov equation, or equivalently, its corresponding gradient flows. Using gradient flows to study the approximation of the optimal transport is well established in the literature, see, for example, the books by Villani, and Ambrosio et al.
>
> •	Writing
>
> o	Some theoretical results seem to be already well-established, e.g., results in Theorem B.1 are all covered by Villani (2008) and Ambrosio et al. (2008). The authors may need to rephrase these results as an introduction to the background knowledge of this area, so that the novel theoretical contributions can be highlighted.
>
> We will revise this part of the paper. We will present the existing results as a proposition and highlight our results as you suggested in the revision.
>
> o	The main paper frequently refers to the appendix, which feels a little disturbing.
>
> We try to balance the readability and technical details, so we put most of the details in the appendix. We also wanted to make sure that the interested reader can find easily find them. That is why we refer to the appendix in the main part of the paper to point out where the details can be found.
>
> o	It seems to me that q(x) in Eq. (2) should be q(z), since q is the density of μ, which represents the easy-to-sample measure/distribution on the latent space, and Z∼μ.
>
> The notation q(x) is correct since x just represents the argument of the density function q.

---

> > ### Comment · AnonReviewer3 · 2020-11-20
> > **Thanks for the explanations, but I still hold the confusion.**
> >
> > Thanks for the further details. I agree with all of the statements, except that "the linearization of the Monge-Ampere equation [...] can be equivalently characterized by gradient flows". Does it mean that there exists a function on distributions whose gradient flow characterizes the equation, or the gradient flow of any function on distributions can characterize the equation? I believe the latter is not true, but the method seems to just use the gradient flow of a specific f-divergence, to approximate the optimal transport flow (this is the geodesic flow on the Wasserstein space) which is the gradient flow of the functional $\mu \mapsto \int \Psi(z) \mu(\mathrm{d}z)$ where $\Psi$ is the solution to the Monge-Ampere equation. $\Psi$ is unknown, but it is not arbitrary and cannot be taken as whatever one wants. It is already determined by the two distributions defining the optimal transport problem. Solving the Monge-Ampere equation is not to just taking the f-divergence as an approximate solution without verification. So I think this approximation is not convincing even in a local/linear sense.
> >
> > That "the gradient flow [of an f-divergence] will provide a map that will push forward the reference distribution to the target distribution" is just because the target distribution is the unique minimizer of the f-divergence. I see it has nothing to do with optimal transport, so I do not agree that "this map is an approximation to the optimal one".
> >
> > Theorem B.1 (i) links a vector field in the support space to a tangent vector in the Wasserstein space; (ii) gives the vector-field-form of the gradient (which is a Wasserstein tangent vector) of $\mathcal{L}[\mu]$ defined in Eq. (B-6); (iii) shows the exponential convergence of the gradient flow if $\mathcal{L}[\mu]$ is strongly convex on the Wasserstein space; (iv) shows the equivalence of the vector-field-form gradient flow expression. None is related to optimal transport or supports approximating the optimal transport flow (the geodesic flow) with the gradient flow of an f-divergence.
> > (Please specify the theorem/page when citing Villani (2008) and Ambrosio et al. (2008) for a specific argument.)
> >
> > * Writing.
> > If $x$ represents the argument of the density function of $q$, then the notation $q(x)$ is correct, but this contradicts the notation in the first paragraph of Section 2, where $z$ is used to represent a sample from distribution $\mu$ whose density function is $q$, and $z$ is fed to $\mathcal{T}$ which is formalized as $\nabla \Psi$ so $z$ should be fed to $\nabla \Psi$. Also, $z$ is the conventional notation for the latent variable from an easy-to-sample distribution in the field of generative models.

---

> > > ### Author Response · Authors · 2020-11-22
> > > **Response to AnonReviewer3: further clarification**
> > >
> > >
> > > We are very grateful to you for clarifying your comments. Now we fully understand your comments and think we are able to address your concern satisfactorily.
> > >
> > > Again, as we stated earlier, we did not claim that we found the optimal map, but we agree that our wording can cause confusion. In "the linearization of the Monge-Ampere equation [...] can be equivalently characterized by gradient flows."  We mean that the linearized equations (B-4) and (B-5) of the Monge-Ampere equation can be characterized by gradient flow by choosing a proper time-dependent $\Psi$, see (ii) in Theorem B.2.  Our linearization of the Monge-Ampere equation with second-order linear elliptic equations follows from Sections 4.1.2-4.1.3 in Villani (2003) (Topics in Optimal Transportation).
> > >
> > > Equation (8.48) in Proposition 8.4.6 of Ambrosio et al. (2008)  shows the connection (locally) of the velocity $v_t$ of the gradient flow $\mu_t$ and the  optimal transport along $\mu_t$, i.e., let  $T_{\mu_t}^{\mu_{t+h}}$ be the optimal transport from $\mu_t$ to $\mu_{t+h}$ for a small $h>0$, then  $T_{\mu_t}^{\mu_{t+h}} = I + h v_t + o(h)$ in  $L^p$, where $I$ is the identity map.  So locally, $I+hv_t$ approximates the optimal transport map from $\mu_t$ to $\mu_{t+h}$ on $[t, t+h]$.  Indeed, we cannot say anything about the global approximation property of the proposed method. This is a challenging problem that needs further study.
> > >
> > > We agree that the sentence "this map is an approximation to the optimal one" can cause confusion. The pushforward map learned via EPT is an approximation to the pushforward map determined by the gradient flow (B-8).  This pushforward map determined by the gradient flow (B-8) pushes the reference distribution to the target distribution, indeed, it may be different from the optimal one. The pushforward map we constructed is motivated by the Monge-Ampere equation that characterizes the optimal map.
> > >
> > > We really appreciate your comments which are very helpful to us in clarifying this important point in the paper.  We will make this point clear in the revision.
> > >
> > > Writing: We agree it is better to use $z$ in the argument of $q$ in this equation. Thanks very much for this comment again. We will make the change in the revision.

---

> > > > ### Comment · AnonReviewer3 · 2020-11-24
> > > > **Thanks for elaborating on this. But the presentation in the updated version still seems be a little misleading.**
> > > >
> > > > Thanks for the detailed explanations. So I suppose the considered and leveraged optimal transport is not the one between the easy-to-sample latent-variable distribution $\mu$ and data distribution $\nu$. Nevertheless, the updated paper still leaves me an impression that the gradient flow solves the Monge-Ampere equation between $\mu$ and $\nu$.
> > > >
> > > > From my perspective, a clearer way to motivate the algorithm may be to find a transport path from $\mu$ to $\nu$ by simulating the gradient flow of the f-divergence between them, which minimizes their difference thus drives $\mu$ to $\nu$. Proposition 8.4.6 of Ambrosio et al. (2008) points out the way for simulating the gradient flow locally, i.e., applying $x \mapsto x + \varepsilon v_t(x)$ to all current samples, where the tangent vector $v_t$ of the gradient flow of the f-divergence is given by Lemma 10.4.1 of Ambrosio et al. (2008). The Monge-Ampere equation does not actually help in this process.

---

> > > > > ### Author Response · Authors · 2020-11-24
> > > > > **To AnonReviewer3: Thanks for the additional comments, but we are afraid you still misunderstand our work**
> > > > >
> > > > > Thank you for the additional comments.  However, based on the comments we are afraid you still misunderstood our work and did not provide a fair assessment. Also, we believe there are multiple reasonable ways to motivate the algorithm.
> > > > >
> > > > > 1.	Ae we have made clear in the revision that our method is motivated by the Monge-Ampere equation. In particular, we highlight and emphasize in the Conclusion section: `` The proposed EPT method is motivated by the Monge-Ampere equation that characterizes the optimal transport map. However, while the EPT map pushes forward a reference distribution to the target, it is not an estimate of the optimal transport map itself. How to consistently estimate the Monge-Ampere optimal map is a challenging and open problem.’’
> > > > >
> > > > > 2.	Our method is based on a first-order approximation to the Monge-Ampere equation. So although the Monge-Ampere equation does not play a direct role in learning the transport map, it is the starting point for us to derive our proposed method. We feel that this connection is interesting and worthwhile to point out.
> > > > >
> > > > > 3.	An important advantage of the proposed approach is that it allows general energy functional in constructing the gradient flow from the reference distribution to the target distribution. In addition to the f-divergence, we can also use the Lebesgue norm of the density difference. Specifically, in the section on Related works, we mentioned that ``MMD flow can be recovered from  EPT  by first choosing $\mathcal{L}[\cdot]$ as the Lebesgue norm  and then projecting the corresponding velocity vector fields onto reproducing kernel Hilbert spaces, please see Appendix B.4 for a proof.’’
> > > > >
> > > > > We think what you suggested is also a reasonable way to motivate the algorithm. We would be very open-minded about this motivation. However, we strongly believe it is not the only way to motivate it. Just like most machine learning problems that have multiple ways to approach them, this generative learning problem also has multiple ways to approach it and multiple solutions.  We believe that in academic research and in this field in particular, it is important to be open-minded about different ways to motivate a solution to a problem.

---

### Official Review · AnonReviewer1 · 2020-11-02
**Technical sound while missing comparison with score-based generative models**

**Rating:** 6
**Confidence:** 4

**Review:**

==== Summary ====

This paper tackles generative modeling (sampling, in particular) via finding the push forward functions T (equivalently, the velocity fields v) that iteratively moves particles from a reference distribution toward the target data distribution. The velocity fields are solved by minimizing the f-divergence between the particle density at iteration k and the target data density, which is shown to be in the form of gradient of density ratio. Based on this intuition, the training stage becomes estimating the density ratio via neural networks, for each iteration k=1,...,K. However, estimating the density ratio can be quite difficult when two densities have little overlapping support. Thus, the author proposes to add gradient regularizer to the density ratio estimating function. The experiment on real-world computer vision benchmarks demonstrate reasonable sampling quality, and the FID score on CIFAR-10 is comparable to some GAN baselines in the generative modeling literature.


==== Pros and Cons ====

Pros:

(1) Technical sounds and interesting insights for non-parametric density ratio estimation with low-dimensional assumptions.

(2) Writing is clear, sections/paragraphs are well structured.

Cons:

(1) The training and inference time seems way larger than GANs because of the iterative updating nature. The inference time is more close to score-based generative models.

(2) Finding velocity fields and conduct inference is very related to score-based generative modeling [1,2,3], and the author should discuss the connections, see detailed below.

(3) The experiments on real-world datasets are not very compelling, and lack some ablation studies, see detailed below.


==== Technical Questions ====

Q1:What are the connections to score-based generative models? For example, [1] also analyze the particle evolution with the Vlasov process, and interpreting it as a gradient flow for minimizing the KL divergence. [2,3,4] are state-of-the-art score-based generative models that iteratively move particles based on the velocity field (i.e., score functions). The experimental comparison is missing.

Q2: The learning objective of decoder G_theta is not justified. Why squared loss? Is that just memorizing the Y_tilde?

Q3: Lack of ablation study for real-world datasets. What is the FID score in CIFAR-10 for your method without the outer loop in Algo 2?  What is the number of inner/outer iterations? Also from the Table A3 and A4, it seems like the coefficient for gradient regularizer alpha = 0. Does this mean that the density ratio estimation problem does not suffer from the no overlapping support issue? If so, why introducing it in Section 4?

Q4: What are the training and inference time for high-dimensional image datasets such as CIFAR-10 and CelebA? In Algorithm 2, how many density ratio estimators R_{phi}^{k} do you learn? Is it that for each outer-inner loop (j,k) pair, you need to learn one? If so, that seems like a lot of models.

==== Additional Feedback / minor suggestions ====

(1) Subplots in Figure 3 are too small. I can not see the figures’ legend.

(2) The Pytorch code of EPT link is not working (page not found on github).

(3) In the first paragraph of Section 3, the right-hand-side of velocity fields seems to have unbalanced parenthesis: f’’(R)(x)) => f’’(R(x)) ?

==== Reference ====

[1] Stein Variational Gradient Descent as Gradient Flow, NeurIPS 2017

[2] Generative Modeling by Estimating Gradients of the Data Distribution, NeurIPS 2019

[3] Improved Techniques for Training Score-Based Generative Models, NeurIPS 2020

[4] Denoising Diffusion Probabilistic Models, NeurIPS 2020

---

> ### Author Response · Authors · 2020-11-16
> **Response to AnonReviewer1**
>
> We are grateful to you for taking the time to review our paper. We appreciate your noticing the merits of our work.
> Below are our responses to your questions and comments.
>
> Q1. SVGD in [1] and our EPT are both particle methods based on gradient flow in measure spaces. However, the SVGD is analyzed in the context of sampling from an un-normalized density, while, our EPT focuses on generative learning, i.e., learning the distribution from samples.  At the population level, projecting the velocity fields of EPT with KL divergence onto reproducing kernel Hilbert Spaces will recover the velocity fields of SVGD. We will add proof of this in the appendix in the revision.
> Score-based methods in [2-4] are also particle methods based on unadjusted Langevin flow and deep score estimator. At the population level, the velocity fields of score-based methods in [2-4] are random since it has a Brownian motion term, while the velocity fields of EPT is deterministic. At the sample level, score-based methods in [2-4] need to employ a deep score estimator which is a vector-valued function, while in EPT we need to estimate the density ratios which are scalar functions. We will add an experimental comparison between our EPT method and score-based generative models.
>
> Q2. The inner loop of EPT is a particle method, i.e., the inner loop of EPT will push forward the initial particle $G_{\theta}(Z_i)$ to samples $\tilde{Y_i}$ that are distributed according to the target distribution. (If no outer loop we have to run the inner loop again if we want to have new samples.) Then we will update the decoder $G_{\theta}$ via minimizing least-squares loss. Because of the paired relationship of $Z_i$ and $\tilde{Y_i}$, the least-squares loss is utilized to train $G_{\theta}$ that approximates the pushforward map from the reference distribution to the target distribution. Therefore, it is not simply memorizing the data. We have verified this via the latent space interpolation.
>
> Q3. The FID score in CIFAR10 without the outer loop in Algorithm 2 is about 64. The number of inner loops is set to 20 on CIFAR10 and CelebA. For the outer loop, its number varies for different datasets (e.g., 5000 for CIFAR10). We use simulated datasets to illustrate the effectiveness of the proposed gradient regularizer. For the real image datasets, we set the regularizer coefficient alpha = 0. We will add the ablation experiments on real datasets with varying alphas.
>
> Q4. The training time of EPT without outer loops is shorter than that of GANs.  To make inference faster, we use outer loops to approximate the pushforward distribution by learning an encoder G_theta. With the outer loops, the training and inference time of EPT is close to that of GANs and is shorter than score-based generative models. We will show the comparisons in the appendix in the revision. In Algorithm 2, only one neural network is introduced to gradually estimate the density ratio between the target distribution and the pushforward distribution represented with evolving particles. The network is shared for different outer-inner loop (j,k) pairs which is similar to a shared discriminator in the whole learning procedure of GANs. Thus, the model is actually simple.
>
> ==== Additional Feedback / minor suggestions ====
>
> (1)	Subplots in Figure 3 are too small. I can not see the figures’ legend.
>
> We will redo Figure 3 to make the legend clear.
>
> (2)	The Pytorch code of EPT link is not working (page not found on github).
>
>  The link is not available due to the anonymous requirement.  We will make it available at a later time.
>
> (3)	In the first paragraph of Section 3, the right-hand-side of velocity fields seems to have unbalanced parenthesis: f’’(R)(x)) => f’’(R(x)) ?
>
>   Thanks, we will correct this.

---

### Decision · Program_Chairs · 2021-01-07
**Final Decision**

**Decision:**

Reject

**Comment:**

The paper proposes a discretization of Wasserstein gradient flow with an euler scheme, and propose a way to estimate each step of the euler scheme using ratio estimators from samples regularized with gradient penalties. Statistical bounds are given to bound the estimated flows from the wasserstein flow.

Reviewers have raised concerns regarding the assumptions under which results present in the paper hold, this was clarified by the authors (goedesic lambda convexity, log sobolev constant for the target density . lipchitizity of velocity fields).  The paper needs a revision to incorporate that feedback and to be in shape for publication.

Other concern were on earlier claims in the paper regarding the monge ampere equation and approximation of the optimal mapping this was addressed by the rebuttal.

Other concerns were also on explaining the relation of the work to score based models and energy based models.

Overall the paper needs to state in a clearer way the assumptions for the theoretical results and to acknowledge the limitations of those assumptions in analyzing the euler scheme.